# Rethinking Value Function Learning for Generalization in Reinforcement Learning

**Seungyong Moon**[1,2], **JunYeong Lee**[1,2], **Hyun Oh Song**[1,2,3*]
[1]Seoul National University, [2]Neural Processing Research Center, [3]DeepMetrics
{symoon11,mascheroni99,hyunoh}@mllab.snu.ac.kr

## Abstract

Our work focuses on training RL agents on multiple visually diverse environments to improve observational generalization performance. In prior methods, policy and value networks are separately optimized using a disjoint network architecture to avoid interference and obtain a more accurate value function. We identify that a value network in the multi-environment setting is more challenging to optimize and prone to memorizing the training data than in the conventional single-environment setting. In addition, we find that appropriate regularization on the value network is necessary to improve both training and test performance. To this end, we propose Delayed-Critic Policy Gradient (DCPG), a policy gradient algorithm that implicitly penalizes value estimates by optimizing the value network less frequently with more training data than the policy network. This can be implemented using a single unified network architecture. Furthermore, we introduce a simple self-supervised task that learns the forward and inverse dynamics of environments using a single discriminator, which can be jointly optimized with the value network. Our proposed algorithms significantly improve observational generalization performance and sample efficiency on the Procgen Benchmark.

## 1   Introduction

In recent years, deep reinforcement learning (RL) has achieved remarkable success in various domains, such as robotic controls and games [27, 19, 32]. To apply RL algorithms to more practical scenarios, such as autonomous vehicles or healthcare systems, they should be robust against the non-stationarity of real-world environments and capable of performing well on unseen situations during deployment. However, current state-of-the-art RL algorithms often fail to generalize to unseen test environments with visual variations, even if they achieve high performance in their training environments [14, 41, 9]. This problem is referred to as *observational overfitting* [36].

Training RL agents on a finite number of visually diverse environments and testing them on unseen environments is the standard protocol for evaluating observational generalization in RL [10]. Several methods have attempted to improve generalization in this framework by adopting the regularization techniques that originate from supervised learning or training robust state representations via self-supervised learning [9, 20, 30, 26]. However, these methods have mainly focused on developing new auxiliary objectives on the existing RL algorithms intended for the conventional single-environment setting such as PPO [34]. Some recent works have investigated the interference between policy and value function optimization arising from the multiple training environments and proposed new training schemes that decouple the policy and value network training with a separate network architecture to obtain an accurate value function [11, 29].

---

*Corresponding author

In this paper, we argue that learning an accurate value function on multiple training environments is more challenging than on a single training environment and requires sufficient regularization. We demonstrate that a value network trained on multiple environments is more likely to memorize the training data and cannot generalize to unvisited states within the training environments, which can be detrimental to not only training performance but also test performance on unseen environments. In addition, we find that regularization techniques that penalize large estimates of the value network, originally developed for preventing memorization in the single-environment setting, are also beneficial for improving both training and test performance in the multi-environment setting. However, this benefit comes at the cost of premature convergence, which hinders further performance enhancement.

To address this, we propose a new model-free policy gradient algorithm named *Delayed-Critic Policy Gradient* (DCPG), which trains the value network with lower update frequency but with more training data than the policy network. We find that the value network with delayed updates suffers less from the memorization problem and significantly improves training and test performance. In addition, we demonstrate that it provides better state representations to the policy network using a single unified network architecture, unlike the prior methods. Moreover, we introduce a simple self-supervised task that learns the forward and inverse dynamics of environments using a single discriminator on top of DCPG. Our algorithms achieve state-of-the-art observational generalization performance and sample efficiency compared to prior model-free methods on the Procgen benchmark [10].

## 2 Preliminaries

### 2.1 Observational Generalization in RL

We consider a collection of environments $\mathcal{M}$ formulated as Markov Decision Processes (MDPs). Each environment $m \in \mathcal{M}$ is described as a tuple $(\mathcal{S}_m, \mathcal{A}, T_m, r_m, \rho_m, \gamma)$, where $\mathcal{S}_m$ is the image-based state space, $\mathcal{A}$ is the action space shared across all environments, $T_m : \mathcal{S}_m \times \mathcal{A} \to \mathcal{P}(\mathcal{S}_m)$ is the transition function, $r_m : \mathcal{S}_m \times \mathcal{A} \to \mathbb{R}$ is the reward function, $\rho_m$ is the initial state distribution, and $\gamma \in [0, 1]$ is the discount factor. We assume that the state space has visual variations between different environments. While the transition and reward functions are defined as specific to an environment, we assume that they exhibit some common structures across all environments. A policy $\pi : \mathcal{S} \to \mathcal{P}(\mathcal{A})$ is trained on a finite number of training environments $\mathcal{M}_{\text{train}} = \{m_i\}_{i=1}^n$, where $\mathcal{S}$ is the set of all possible states in $\mathcal{M}$. Our goal is to learn a generalizable policy that maximizes the expected return on unseen test environments $\mathcal{M}_{\text{test}} = \mathcal{M} \setminus \mathcal{M}_{\text{train}}$.

In this paper, we utilize the Procgen benchmark as a testbed for observational generalization [10]. It is a collection of 16 video games with high diversity comparable to the ALE benchmark [5]. Each game consists of procedurally generated environment instances with visually different layouts, backgrounds, and game entities (*e.g.*, the spawn locations and times for enemies), also called levels. The standard evaluation protocol on the Procgen benchmark is to train a policy on a finite set of training levels and evaluate its performance on held-out test levels [10].

### 2.2 Proximal Policy Optimization

Proximal Policy Optimization (PPO) is a powerful model-free policy gradient algorithm that learns a policy $\pi_\theta$ and value function $V_\phi$ parameterized by deep neural networks [34]. For training, PPO first collects trajectories $\tau$ using the old policy network $\pi_{\theta_{\text{old}}}$ right before the update. Then, the policy network is trained with the collected trajectories for several epochs to maximize the following clipped surrogate policy objective $J_\pi$ designed to constrain the size of policy update:

$$J_\pi(\theta) = \mathbb{E}_{s_t, a_t \sim \tau} \left[ \min \left( \frac{\pi_\theta(a_t \mid s_t)}{\pi_{\theta_{\text{old}}}(a_t \mid s_t)} \hat{A}_t, \ \text{clip} \left( \frac{\pi_\theta(a_t \mid s_t)}{\pi_{\theta_{\text{old}}}(a_t \mid s_t)}, 1 - \epsilon, 1 + \epsilon \right) \hat{A}_t \right) \right],$$

where $\hat{A}_t$ is an estimate of the advantage function at timestep $t$. Concurrently, the value network is trained with the collected trajectories to minimize the following value objective $J_V$:

$$J_V(\phi) = \mathbb{E}_{s_t \sim \tau} \left[ \frac{1}{2} \left( V_\phi(s_t) - \hat{R}_t \right)^2 \right],$$

where $\hat{R}_t = \hat{A}_t + V_\phi(s_t)$ is the value function target. It is used to compute the advantage estimates via generalized advantage estimator (GAE) [33].

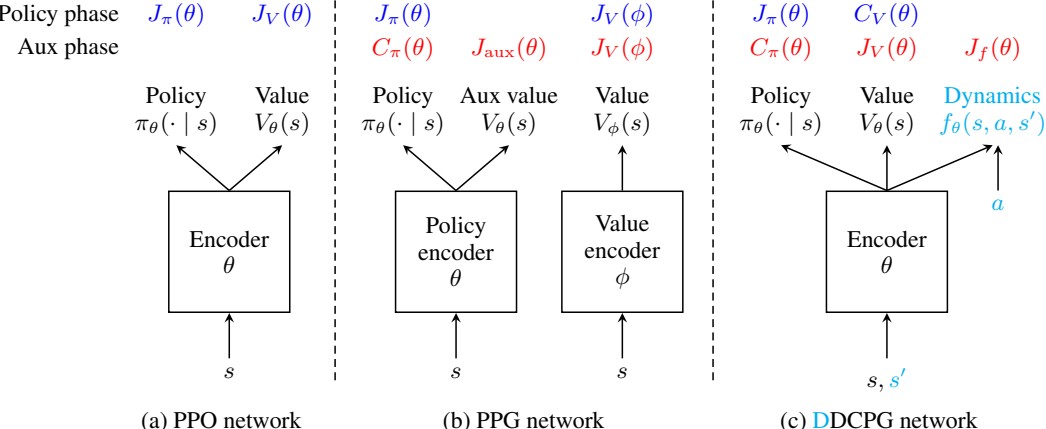

Figure 1: Network architectures for PPO, PPG, and DDCPG. The objectives $J_\pi$, $J_V$, $J_{\text{aux}}$, and $J_f$ denote the policy, value, auxiliary value, and dynamics objectives, respectively. The regularizers $C_\pi$ and $C_V$ denote the policy and value regularizers, respectively. The blue and red terms represent optimization problems during the policy and auxiliary phases, respectively.

In practice, the policy and value networks are jointly optimized with shared parameters (*i.e.*, $\theta = \phi$), especially in image-based RL [13, 39]. For example, they can be implemented using a shared encoder followed by separate linear heads, as shown in Figure 1a. Sharing parameters is advantageous in that representations learned by each objective can be beneficial to the other. It also reduces memory costs and accelerates training time. However, a shared network architecture complicates the optimization as a single encoder should be optimized over multiple objectives whose gradients may have varying scales and directions. It also constrains the policy and value networks to be optimized under the same training hyperparameter setting, such as batch size and the number of epochs, severely limiting the flexibility of PPO.

## 2.3 Phasic Policy Gradient

Phasic Policy Gradient (PPG) is an algorithm built upon PPO that significantly improves observational generalization by addressing the problems of sharing parameters [11]. More specifically, PPG employs separate encoders for the policy and value networks, as shown in Figure 1b. In addition, it introduces an auxiliary value head $V_\theta$ on top of the policy encoder in order to distill useful representations from the value network into the encoder. For training, PPG alternates between policy and auxiliary phases. During the policy phase, which is repeated $N_\pi$ times, the policy and value networks are trained with newly-collected trajectories to optimize the policy and value objectives from PPO, respectively. Then, all states and value function targets in the trajectories are stored in a buffer $\mathcal{B}$. During the auxiliary phase, the auxiliary value head and the policy network are jointly trained with all data in the buffer to optimize the following auxiliary value objective $J_{\text{aux}}$ and policy regularizer $C_\pi$:

$$J_{\text{aux}}(\theta) = \mathbb{E}_{s_t \sim \mathcal{B}} \left[ \frac{1}{2} \left( V_\theta(s_t) - \hat{R}_t \right)^2 \right], \quad C_\pi(\theta) = \mathbb{E}_{s_t \sim \mathcal{B}} \left[ D_{\text{KL}}(\pi_{\theta_{\text{old}}}(\cdot \mid s_t) \parallel \pi_\theta(\cdot \mid s_t)) \right],$$

where $\pi_{\theta_{\text{old}}}$ is the policy network right before the auxiliary phase and $D_{\text{KL}}$ denotes the KL divergence. In other words, the value network is distilled into the policy encoder while maintaining the outputs of the policy network unchanged. Moreover, the value network is additionally trained with all data in the buffer to optimize the value objective from PPO to obtain a more accurate value function. It is worth noting that the training data size in the auxiliary phase is $N_\pi$ times larger than the policy phase. It has been claimed that the distillation of a better-trained value network with a separate architecture and the additional training for a more accurate value network can improve observational generalization performance and sample efficiency [11].

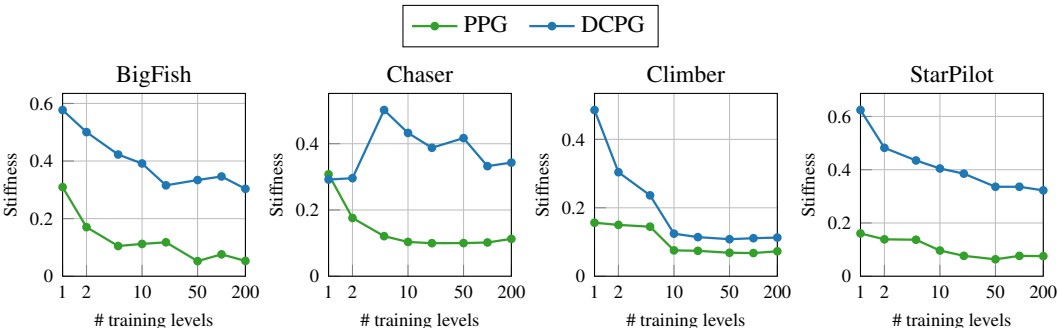

Figure 2: Average stiffness of value networks for PPG and DCPG on 4 Procgen games while varying the number of training levels.

## 3 Motivation

### 3.1 Difficulty of Training Value Network on Multiple Training Environments

We begin by investigating the difficulty of obtaining an accurate value network across multiple training environments. Indeed, learning a value network that better approximates the true value function on the given training environments can result in improved training performance [37]. However, even in a simple setting where an agent is trained on a single environment, it has been shown that a value network is likely to memorize the training data and unable to extrapolate well to unseen states even in the same training environment [23, 21, 12, 15]. This problem can be exacerbated when the number of training environments increases. Intuitively, given the fixed number of environment steps, the value network will be provided fewer training samples per environment and rely more on memorization.

To corroborate this claim, we measure the stiffness of the value network between states $(s, s')$ while varying the number of training environments [16, 6], which is defined by

$$\rho(s, s') = \frac{\nabla_\phi J_V(\phi; s)^\intercal \nabla_\phi J_V(\phi; s')}{\|\nabla_\phi J_V(\phi; s)\|_2 \|\nabla_\phi J_V(\phi; s')\|_2}.$$

Low stiffness indicates that updating the network parameters toward minimizing the value objective for one state will have a negative effect on the minimization of the value objective for other states [6]. That is, the value network is less able to adjust its parameter to predict the true value function across different states and instead tends to memorize only the states it has encountered. More specifically, we train PPG agents on the Procgen games while increasing the number of training levels from 0 to 200 and compute the average stiffness across all state pairs in a mini-batch of size $2^{14}$ (=16,384) throughout training. The detailed experimental settings and results can be found in Appendix A.

The green lines in Figure 2 show that the stiffness of the value network decreases as the number of training environments increases, as expected. It implies that the value network trained on multiple environments is more likely to memorize the training data and cannot accurately predict the values of unvisited states from the training environments. This memorization problem brings us to train a value network with sufficient regularization.

### 3.2 Training Value Network with Explicit Regularization

Next, we examine the effectiveness of value network regularization in the multi-environment setting. We consider applying two existing regularization techniques developed to prevent the memorization problem in the single-environment setting, especially when training data is limited. The first method is discount regularization (DR), which trains a myopic value network with a lower discount factor $\gamma'$ [28]. The second method is activation regularization (AR), which optimizes a value network with $L_2$ penalty on its outputs:

$$J_V^{\text{reg}}(\phi) = \mathbb{E}_{s_t \sim \tau} \left[ \frac{1}{2} \left( V_\phi(s_t) - \hat{R}_t \right)^2 + \frac{\alpha}{2} V_\phi(s_t)^2 \right],$$

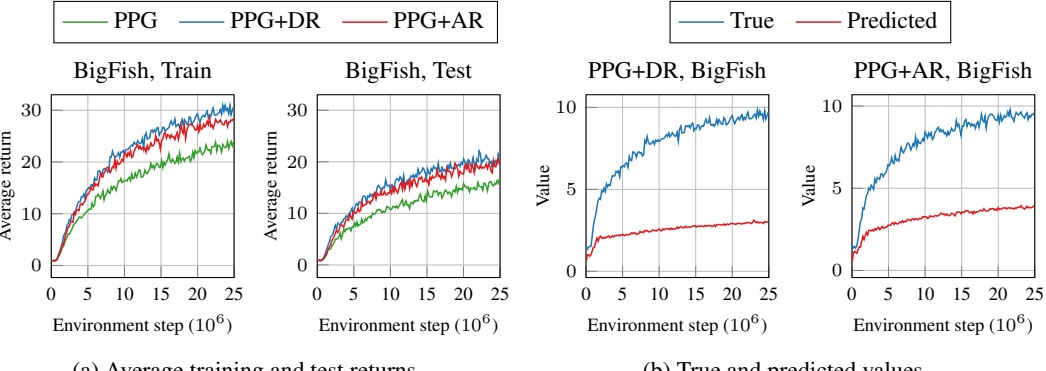

(a) Average training and test returns.    (b) True and predicted values.

Figure 3: (a) Training and test performance curves of PPG, PPG+DR, and PPG+AR on BigFish. (b) True and predicted values measured at the initial states of training environments for PPG+DR and PPG+AR on BigFish. The mean is computed over 10 different runs.

where $\alpha > 0$ is the regularization coefficient [3]. We train PPG agents with each of these two methods using 200 training levels on the Procgen games. We reduce the discount factor from $\gamma = 0.999$ to $\gamma' = 0.995$ for PPG+DR and use $\alpha = 0.05$ for PPG+AR. We measure the average training and test returns to evaluate the training performance and its transferability to unseen test environments.

As shown in Figure 3a, the value network regularization improves the training and test performance of PPG on BigFish to some extent. It implies that explicitly suppressing the value network also helps to mitigate memorization in the multi-environment setting. We also observe that these regularization methods improve the training and test performance across all Procgen games on average. The detailed experimental setting and results can be found in Appendix B.

Despite its effectiveness, explicit value network regularization can lead to a suboptimal solution as the number of environment steps increases. Figure 3b shows the true and predicted values measured at the initial states of the training environments for PPG+DR and PPG+AR on BigFish. The predicted values with explicit regularization reach a plateau too quickly, suggesting that excessive regularization later hinders the value network from learning an accurate value function. This motivates us to develop a more flexible regularization method that boosts training and test performance while allowing the value network to converge to true values.

## 4   Delayed-Critic Policy Gradient

In this section, we present a novel model-free policy gradient algorithm called *Delayed-Critic Policy Gradient* (DCPG), which effectively addresses the memorization problem of the value network in a simple and flexible manner. The key idea is that the value network should be optimized with a larger amount of training data to avoid memorizing a small number of recently visited states, based on the stiffness analysis in Section 3.1. Furthermore, the value network should be optimized with a delay compared to the policy network to implicitly suppress the value estimate, based on the regularization analysis in Section 3.2.

### 4.1   Algorithm

DCPG follows a similar procedure to PPG by alternating policy and auxiliary phases. Still, it employs a shared network architecture in the same way as PPO and does not require any additional auxiliary head, as shown in Figure 1c. During the policy phase, which occurs more frequently but with less training data than the auxiliary phase, the policy network is trained with newly-collected trajectories to optimize the policy objective from PPO. In contrast, the value network is constrained to preserve its outputs by optimizing the following value regularizer $C_V$:

$$C_V(\theta) = \mathbb{E}_{s_t \sim \tau} \left[ \frac{1}{2} \left( V_\theta(s_t) - V_{\theta_{\text{old}}}(s_t) \right)^2 \right],$$

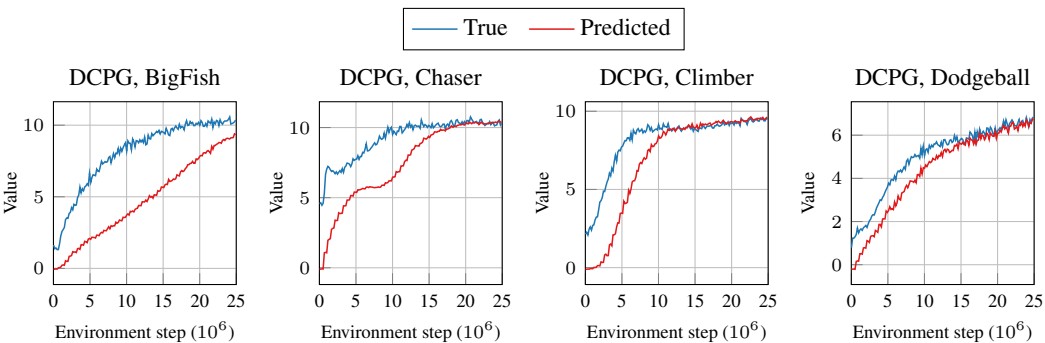

Figure 4: True and predicted values measured at the initial states of training environments for DCPG agents on 4 Procgen games. The mean is computed over 10 different runs.

where $V_{\theta_{\mathrm{old}}}$ is the old value network right before the policy phase. Then, all states and value function targets in the trajectories are stored in a buffer.

During the auxiliary phase, the value network is trained with a larger number of data in the buffer to optimize the value objective from PPO. In contrast, the policy network is constrained to preserve its outputs by optimizing the policy regularizer. Note that since the value network shares the same encoder as the policy network, optimizing the value objective directly plays the role of representation learning for the policy network. Thus, unlike PPG, DCPG does not require an additional auxiliary head for feature distillation. The overall procedure of DCPG can be found in Algorithm 1.

---

**Algorithm 1** Dynamics-aware Delayed-Critic Policy Gradient (DDCPG)

---

**Require:** Policy network $\pi_\theta$, value network $V_\theta$, dynamics discriminator $f_\theta$
1: **for** phase = $1, 2, \ldots$ **do**
2:     Initialize buffer $\mathcal{B}$
3:     **for** iter = $1, 2, \ldots, N_\pi$ **do**                                        ▷ Policy phase
4:         Sample trajectories $\tau$ using $\pi_\theta$ and compute value function target $\hat{R}_t$ for each state $s_t \in \tau$
5:         **for** epoch = $1, 2, \ldots, E_\pi$ **do**
6:             Optimize policy objective $J_\pi(\theta)$ and value regularizer $C_V(\theta)$ with $\tau$
7:         **end for**
8:         Add $(s_t, a_t, \hat{R}_t)$ to $\mathcal{B}$
9:     **end for**
10:    **for** iter = $1, 2, \ldots, E_{\mathrm{aux}}$ **do**                                ▷ Auxiliary phase
11:       Optimize value objective $J_V(\theta)$, dynamics objective $J_f(\theta)$, and policy regularizer $C_\pi(\theta)$ with $\mathcal{B}$
12:    **end for**
13: **end for**

---

### 4.2 Value Network Analysis of Delayed Critic Update

The delayed critic update in DCPG can function as implicit regularization of the value network. Since the policy improvement is not immediately reflected in the value network, it encourages smaller value estimates than the true values. To validate this claim, we train DCPG agents using 200 training levels on the Procgen games and measure the true and predicted values at the initial states of the training levels. Figure 4 shows the value estimates are kept lower than the true values at the early stage of training but recover the true values as training progresses. More results can be found in Appendix C.

We also evaluate the effectiveness of the delayed critic update in mitigating the memorization problem by measuring the stiffness of the value network as done in Section 3.1. The blue lines in Figure 2 show that DCPG has higher stiffness than PPG, implying that the value network of DCPG is more robust against memorization. We conclude that the delayed critic update serves as an effective regularizer for the value network. More results can be found in Appendix A.

### 4.3 Learning Forward and Inverse Dynamics with Single Discriminator

In the context of multi-task learning, learning an auxiliary task can operate as a good regularizer that prevents memorization in the main task [31]. Motivated by this, we consider learning an auxiliary task using the representations from the encoder and jointly learning the value network with the auxiliary objective. Suppose the buffer $\mathcal{B}$ used in the auxiliary phase contains a transition tuple $(s_t, a_t, s_{t+1})$. Since the transition function from each environment is assumed to have the same underlying structure, it is natural to train a neural network that models the forward dynamics of environments in the latent space of the encoder. Concretely, a dynamics head $f_\theta$ takes the embedding of the current state $s_t$ and the current action $a_t$ as inputs and predicts the embedding of the next state $s_t$. However, it is well known that a forward dynamics model using a neural network tends to overfit a small number of training data and make poor predictions on unseen states [8]. To address this, we consider learning the forward dynamics by training a discriminator that determines whether the next state is valid, given the current state and action:

$$f_\theta(s_t, a_t, s_{t+1}) = 1, \ f_\theta(s_t, a_t, \hat{s}_{t+1}) = 0, \ \hat{s}_{t+1} \in \mathcal{B} \setminus \{s_{t+1}\}.$$

Such a discriminator might discard information about the action to distinguish whether a transition is valid. For example, the discriminator can predict the valid transition by using only the proximity of the current and next states (see Appendix F for more details). To avoid this, we train the discriminator to jointly distinguish whether the current action is valid, given the current and next states (*i.e.*, inverse dynamics):

$$f_\theta(s_t, a_t, s_{t+1}) = 1, \ f_\theta(s_t, \hat{a}_t, s_{t+1}) = 0, \ \hat{a}_t \in \mathcal{B} \setminus \{a_t\}.$$

Finally, we optimize the following dynamics objective during the auxiliary phase:

$$J_f(\theta) = \mathbb{E}_{s_t, a_t, s_{t+1} \sim \mathcal{B}} \big[ \log\left(f_\theta(s_t, a_t, s_{t+1})\right) + \log\left(1 - f_\theta(s_t, a_t, \hat{s}_{t+1})\right)$$
$$+ \eta \log\left(1 - f_\theta(s_t, \hat{a}_t, s_{t+1})\right) \big],$$

where $\hat{s}_{t+1} \sim \mathcal{U}(\mathcal{B} \setminus \{s_{t+1}\})$, $\hat{a}_t \sim \mathcal{U}(\mathcal{B} \setminus \{a_t\})$, and $\eta > 0$ is the coefficient for the inverse dynamics. We name the resulting algorithm *Dynamics-aware Delayed-Critic Policy Gradient* (DDCPG). The network architecture and algorithm for DDCPG are shown in Figure 1c and Algorithm 1, respectively (the differences with DCPG are marked in cyan).

## 5 Experiments

### 5.1 Experimental Settings

We evaluate the observational generalization performance of our methods on the Procgen benchmark. We use the "easy" difficulty mode, which most prior works have focused on. We train agents on 200 training levels generated by seeds from 0 to 200 for 25M environment steps and test them on held-out test levels, following the practice in Cobbe et al. [10]. We measure the average return of 100 test episodes and report its mean and standard deviation over 10 runs with different initialization.

We compare our methods with PPO and 4 other baselines that use PPO as the backbone algorithm: UCB-DrAC, PLR, PPG, and DAAC [30, 22, 11, 29]. UCB-DrAC is a data augmentation algorithm for training policy and value networks to be robust against various transformations. PLR is a sampling algorithm for the procedural generation that selects the next training level based on its future learning potential. PPG is the previous state-of-the-art method on the Procgen benchmark. DAAC is motivated by PPG and distills the advantage function into the policy network instead. For each method, we use the same set of hyperparameters across all games, following the standard protocol in the ALE and Procgen benchmarks [27, 10]. For our methods, we use the same hyperparameter setting as PPG for a fair comparison. To compare the performance of each method with a single score, we calculate the PPO-normalized score averaged over all Procgen games, which is computed by dividing the average return of each method by PPO, following the practice in Jiang et al. [22], Raileanu et al. [30]. For the implementation details and hyperparameters, please refer to Appendix D. The code can be found at `https://github.com/snu-mllab/DCPG`.

### 5.2 Observational Generalization Performance on Procgen Benchmark

Table 1 shows the average test returns of each method on all 16 Procgen games. DCPG significantly outperforms all the baselines, yielding a 24%p improvement on average compared to the previous

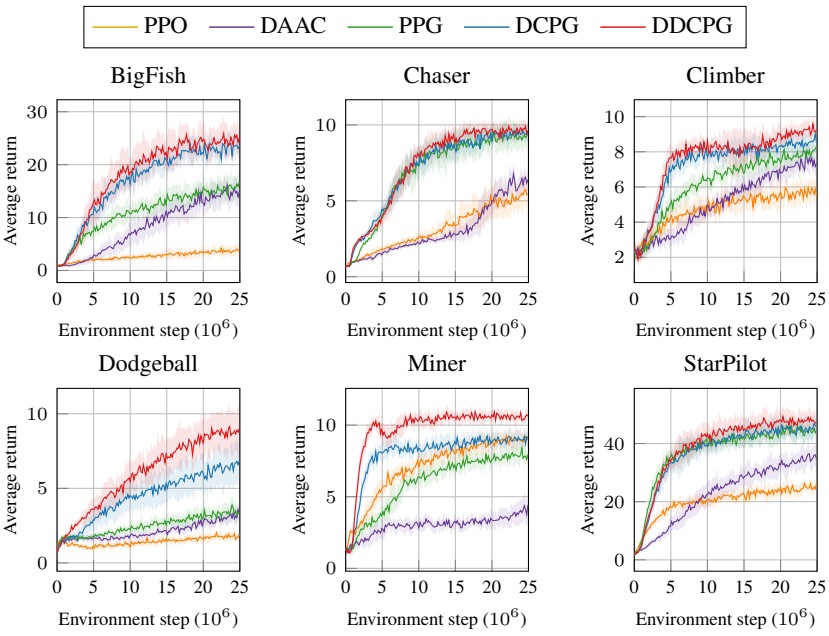

Figure 5: Test performance curves of each method on 6 Procgen games. Each agent is trained on 200 training levels for 25M environment steps and evaluated on 100 unseen test levels. The mean and standard deviation are computed over 10 different runs.

Table 1: Average test returns of each method on all 16 Procgen games. Each agent is trained on 200 training levels for 25M environment steps and evaluated on 100 unseen test levels. The mean and standard deviation are computed over 10 different runs.

| Environment | PPO | UCB-DrAC | PLR | DAAC[2] | PPG | DCPG | DDCPG |
|---|---|---|---|---|---|---|---|
| BigFish | $3.4 \pm 1.0$ | $6.6 \pm 2.5$ | $10.8 \pm 2.5$ | $15.3 \pm 3.0$ | $16.4 \pm 2.0$ | $23.1 \pm 2.3$ | $25.9 \pm 2.3$ |
| BossFight | $7.4 \pm 0.7$ | $7.2 \pm 1.0$ | $8.8 \pm 0.7$ | $9.5 \pm 0.8$ | $10.4 \pm 0.6$ | $10.2 \pm 0.4$ | $10.6 \pm 0.5$ |
| CaveFlyer | $5.2 \pm 0.6$ | $4.4 \pm 0.8$ | $6.3 \pm 0.7$ | $5.1 \pm 0.6$ | $7.7 \pm 0.7$ | $6.3 \pm 0.4$ | $6.0 \pm 0.3$ |
| Chaser | $5.5 \pm 0.8$ | $6.7 \pm 0.5$ | $7.5 \pm 0.8$ | $6.3 \pm 0.8$ | $8.9 \pm 0.9$ | $9.7 \pm 0.4$ | $9.9 \pm 0.6$ |
| Climber | $5.5 \pm 0.6$ | $6.2 \pm 0.6$ | $6.4 \pm 0.5$ | $7.4 \pm 0.5$ | $8.1 \pm 0.4$ | $8.5 \pm 0.7$ | $9.3 \pm 0.7$ |
| CoinRun | $8.8 \pm 0.3$ | $8.6 \pm 0.4$ | $8.9 \pm 0.3$ | $9.3 \pm 0.2$ | $8.9 \pm 0.3$ | $8.5 \pm 0.4$ | $8.4 \pm 0.3$ |
| Dodgeball | $1.9 \pm 0.3$ | $4.5 \pm 1.4$ | $2.2 \pm 0.4$ | $3.2 \pm 0.5$ | $3.6 \pm 0.7$ | $6.7 \pm 0.4$ | $9.0 \pm 1.6$ |
| FruitBot | $27.5 \pm 1.5$ | $26.9 \pm 1.7$ | $27.7 \pm 1.3$ | $28.5 \pm 1.4$ | $29.1 \pm 1.0$ | $29.0 \pm 1.2$ | $28.8 \pm 0.6$ |
| Heist | $2.6 \pm 0.6$ | $3.4 \pm 1.0$ | $3.1 \pm 0.6$ | $3.0 \pm 0.5$ | $2.8 \pm 0.5$ | $3.3 \pm 0.5$ | $4.4 \pm 0.8$ |
| Jumper | $5.7 \pm 0.4$ | $5.6 \pm 0.4$ | $5.9 \pm 0.4$ | $5.8 \pm 0.4$ | $6.0 \pm 0.4$ | $6.2 \pm 0.5$ | $6.4 \pm 0.4$ |
| Leaper | $5.6 \pm 1.5$ | $3.6 \pm 0.7$ | $7.3 \pm 0.2$ | $8.2 \pm 1.1$ | $7.2 \pm 2.4$ | $6.9 \pm 1.7$ | $6.8 \pm 1.5$ |
| Maze | $5.3 \pm 0.7$ | $5.7 \pm 1.1$ | $5.6 \pm 0.5$ | $5.6 \pm 0.6$ | $5.3 \pm 0.4$ | $5.7 \pm 0.7$ | $6.5 \pm 0.6$ |
| Miner | $9.2 \pm 0.6$ | $10.1 \pm 0.7$ | $9.4 \pm 0.7$ | $4.4 \pm 1.1$ | $7.6 \pm 0.7$ | $9.1 \pm 0.8$ | $10.6 \pm 0.5$ |
| Ninja | $5.7 \pm 0.5$ | $5.7 \pm 0.5$ | $7.1 \pm 0.5$ | $6.7 \pm 0.7$ | $6.7 \pm 0.4$ | $6.3 \pm 0.6$ | $6.6 \pm 0.5$ |
| Plunder | $5.0 \pm 0.5$ | $8.1 \pm 1.5$ | $8.7 \pm 1.3$ | $5.6 \pm 0.6$ | $13.7 \pm 1.7$ | $13.7 \pm 1.3$ | $12.9 \pm 2.5$ |
| StarPilot | $25.8 \pm 2.0$ | $27.0 \pm 3.2$ | $24.9 \pm 3.6$ | $35.9 \pm 3.3$ | $44.8 \pm 2.6$ | $46.1 \pm 2.1$ | $46.6 \pm 4.1$ |
| PPO-norm score (%) | $100.0 \pm 3.1$ | $120.7 \pm 10.7$ | $130.0 \pm 6.0$ | $136.7 \pm 7.6$ | $160.3 \pm 6.3$ | $\mathbf{184.5 \pm 5.2}$ | $\mathbf{202.2 \pm 10.2}$ |

state-of-the-art PPG. Adding dynamics learning to DCPG can further improve the test performance with an 18%p increase in the PPO-normalized score. We also provide the average training returns on all Procgen games in Appendix E, showing that our methods achieve better sample efficiency in the multi-environment setting. Furthermore, we evaluate the performance of our methods using the Min-Max normalized score and report mean, median, and interquartile mean (IQM) scores in Appendix G, following the recent practice in Agarwal et al. [2]. We observe that the performance improvements of our methods are statistically significant for all the metrics we consider.

Figure 5 shows the test performance curves of each method on 6 Procgen games throughout training. Our methods achieve superior final test performance and acquire generalization ability with fewer environment interactions. For example, our methods reach the final performance of PPG using only

---

[2]Results reported in the original paper use a different hyperparameter setting for each game.

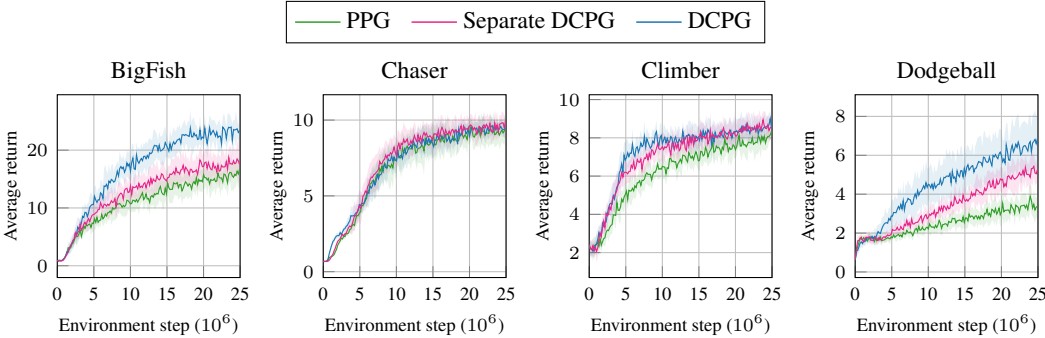

Figure 6: Test performance curves of PPG, Separate DCPG, and DCPG on 4 Procgen games. Each agent is trained on 200 training levels for 25M environment steps and evaluated on 100 unseen test levels. The mean and standard deviation are computed over 10 different runs.

20% of total environment steps on BigFish and Climber and 10% on Dodgeball and Miner. More experimental results can be found in Appendix E.

### 5.3 Ablation Study for Delayed Critic Update

The delayed critic update not only alleviates the memorization problem but also provides better state representations for generalization to the policy network. To validate this, we train DCPG agents with an additional value network $V_\phi$ using a separate encoder (Separate DCPG). The separate value network $V_\phi$ is aggressively optimized in the same way as PPG and only used for calculating advantages in the policy objective. The original value network $V_\theta$ is only used for learning representations for the policy network with delayed updates. Figure 6 shows that Separate DCPG achieves better test performance than PPG, implying that the delayed critic update produces more generalizable representations. For the detailed experimental settings and results, please refer to Appendix F.

### 5.4 Ablation Study for Dynamics Learning

We conduct an ablation study to show the effectiveness of training forward and inverse dynamics using a single discriminator. We train DCPG agents with only forward dynamics learning (DCPG+F) and only inverse dynamics learning (DCPG+I). In addition, we train DCPG agents with forward and inverse dynamics learning using two separate discriminators (DCPG+FI). Table 2 shows that DDCPG, which jointly optimizes the forward and inverse dynamics with a single discriminator, achieves better test performance than the other methods with dynamics learning. Note that inverse dynamics provides performance gain only when jointly trained with forward dynamics using a single discriminator. We provide the detailed experimental settings and results in Appendix F.

Table 2: PPO-normalized score of each method with dynamics learning on all 16 Procgen games. Each agent is trained on 200 training levels for 25M environment steps and evaluated on 100 unseen test levels. The mean and standard deviation are computed over 10 different runs.

|  | DCPG | DCPG+F | DCPG+I | DCPG+FI | DDCPG |
|---|---|---|---|---|---|
| PPO-norm score (%) | $184.5 \pm 5.2$ | $195.9 \pm 7.7$ | $185.5 \pm 6.5$ | $194.6 \pm 6.2$ | $\mathbf{202.2 \pm 10.2}$ |

## 6 Related Works

**Observational generalization in RL** One prominent approach for improving observational generalization in model-free RL is to employ regularization techniques developed for supervised learning, such as batch normalization, weight regularization, and information bottleneck [14, 9, 20]. Several works adopt data augmentation strategies commonly used in computer vision to address the generalization problem [24, 38]. For example, UCB-DrAC proposes a method that automatically finds an effective augmentation and introduces new regularization terms to learn robust state representations

under various image transformations [30]. Recently, it has been shown that MuZero Reanalyse, the state-of-the-art model-based RL algorithm, achieves better observational generalization performance than model-free approaches, though it requires a larger network architecture [4].

Another line of work uses self-supervised learning to learn invariant representations that can generalize to unseen environments. Some methods attempt to learn state representations that capture long-term behavior proximity between states using bisimulation or policy similarity metrics to discard irrelevant information for generalization [40, 1, 26]. The idea of learning dynamics as a self-supervised task, originally developed for sample efficiency, has also been used to improve observational generalization [18, 35]. DIM learns a state representation that can predict the representations of successive timesteps using self-supervised learning [25]. It has also been tested that predicting future state representations conditioned on the current state and action improves the generalization performance of MuZero [4]. Our work aims to enhance dynamics learning further by jointly modeling the forward and inverse dynamics objectives using a single discriminator.

PPG is most closely related to our work, which proposes a better training algorithm to improve both observational generalization and sample efficiency in the multi-environment setting [11]. It trains the policy and value networks independently using a separate architecture and different levels of sample reuse to avoid interference and obtain a better-trained value network. DAAC also uses decoupled policy and value networks for a more accurate value function but distills the advantage function into the policy encoder under the assumption that it is less prone to overfitting to environment-specific features than the value function [29]. In contrast, we focus on regularizing the value network while obtaining better representation, which distinguishes our work from the prior methods.

**Learning value networks with regularization**   Discount regularization is one common regularization method to mitigate overfitting in the value network [7]. A lower discount factor can lead to better performance in approximated dynamic programming when the approximation error is large [28]. It is also helpful for preventing overfitting when dealing with limited data in POMDPs [17]. Activation regularization penalizes the outputs of the value network and has a regularization effect similar to that of discount regularization [3]. It has been shown that activation regularization is equivalent to discount regularization in batch TD learning and also effective in online RL [3].

# 7   Conclusion

We have investigated the difficulty of learning an accurate value network in the multi-environment setting and shown that suppressing the value network with appropriate regularization can be helpful to improve both the training and test performance. Based on this observation, we propose Delayed-Critic Policy Gradient (DCPG), where the value network is implicitly regularized to have lower predicted values at the early stage of training with delayed updates compared to the policy network. We find that the delayed value update prevents the memorization of training data and produces more generalizable representations that can be extended to unseen test environments. Furthermore, we introduce a simple self-supervised task that jointly learns forward and inverse dynamics using a single discriminator, which can be easily combined with DCPG. Our algorithms exhibit state-of-the-art performance in observational generalization on the challenging Procgen benchmark.

# Acknowledgements

This work was supported in part by Samsung Advanced Institute of Technology, Samsung Electronics Co., Ltd., Institute of Information & Communications Technology Planning & Evaluation (IITP) grant funded by the Korea government (MSIT) (No. 2019-0-01371, Development of brain-inspired AI with human-like intelligence, No. 2020-0-00882, (SW STAR LAB) Development of deployable learning intelligence via self-sustainable and trustworthy machine learning, and No. 2022-0-00480, Development of Training and Inference Methods for Goal-Oriented Artificial Intelligence Agents), and a grant of the Korea Health Technology R&D Project through the Korea Health Industry Development Institute (KHIDI), funded by the Ministry of Health & Welfare, Republic of Korea (grant number: HI21C1074). This material is based upon work supported by the Air Force Office of Scientific Research under award number FA2386-22-1-4010. Hyun Oh Song is the corresponding author.

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
