# Supplementary Material for Rethinking Value Function Learning for Generalization in Reinforcement Learning

## A  Stiffness Analysis

To quantify how much the number of training environments affects the memorization of the value network, we train PPG agents using the training levels of seeds from 0 to $n$ for 8M environment steps on all 16 Procgen games and measure the stiffness of the value network between states while varying the number of training environments $n \in \{1, 2, 5, 10, 20, 50, 100, 200\}$. Throughout the training, we sample trajectories from the training levels, which consist of $2^{14}$ (=16,384) states, and compute the individual gradient of the value objective for each state using BackPACK [4]. Then, we calculate the mean stiffness of the value network across all state pairs and report its average computed over all training epochs.

The green lines in Figure 1 demonstrate that the stiffness decreases as the number of training levels increases in most of the Procgen games. This implies that a value network trained on more training environments is more likely to memorize training data and cannot extrapolate values of unseen states from the training environments.

We also conduct stiffness experiments for DCPG agents under the same experimental setting as PPG to evaluate how the delayed critic update mitigates the memorization problem. The blue lines in Figure 1 show that DCPG achieves higher stiffness than PPG for all $n$ in 12 of 16 Procgen games. This suggests that the delayed critic update effectively alleviates the memorization problem.

## B  Training Value Function with Explicit Regularization

We train PPG agents with discount regularization (PPG+DR) and activation regularization (PPG+AR) using 200 training levels for 25M environment steps on all 16 Procgen games and report the PPO-normalized training and test scores. For PPG+DR, we sweep the new discount factor $\gamma'$ in a range of $\{0.98, 0.99, 0.995\}$ and find $\gamma' = 0.995$ performs best. For PPG+AR, we sweep the regularization coefficient $\alpha$ within $\{0.01, 0.05, 0.1\}$ and find $\alpha = 0.05$ works best. Table 1 shows that PPG+DR and PPG+AR improve both the training and test scores of PPG by 10%p.

Table 1: PPO-normalized training and test scores of PPG, PPG+DR, and PPG+AR. Each agent is trained on 200 training levels for 25M environment steps. The mean and standard deviation are computed over 10 different runs.

|  | PPG | PPG+DR | PPG+AR |
|---|---|---|---|
| PPO-norm train score (%) | $136.7 \pm 4.4$ | $146.8 \pm 1.4$ | $146.7 \pm 2.6$ |
| PPO-norm test score (%) | $160.3 \pm 6.3$ | $168.0 \pm 5.8$ | $169.9 \pm 6.9$ |

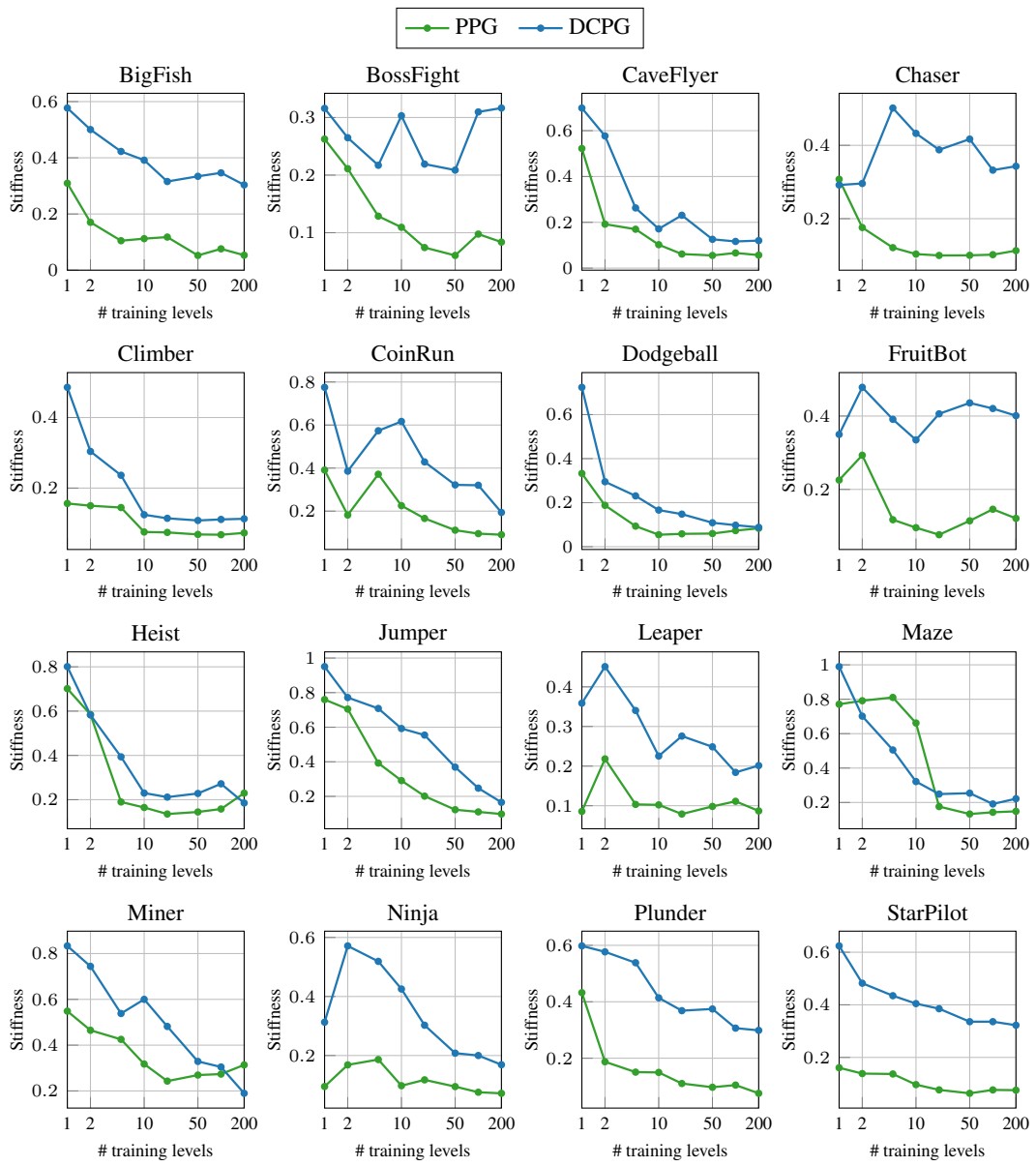

Figure 1: Average stiffness of value networks for PPG and DCPG on all 16 Procgen games while varying the number of training levels. Each agent is trained for 8M environment steps.

## C    Value Network Analysis of Delayed Critic Update

To verify whether the delayed critic update acts as a value network regularizer, we train DCPG agents using 200 training levels for 25M environment steps on all 16 games of Procgen and compare the true and predicted values for the initial states of the training levels. More specifically, we collect 100 training episodes throughout the training and evaluate the value network prediction for the initial state of each trajectory. We estimate the true value of each initial state by computing the discounted return of the corresponding trajectory. We also conduct the same experiments with PPG+DR and PPG+AR for comparison with explicit value function regularization.

The red curves in Figure 2 show that the value network of DCPG consistently underestimates the true values when the number of environment steps is small. In addition, the value predictions of DCPG become less biased as training progresses and reach the true values in most games. It implies that the

delayed value update can serve as implicit regularization that slowly diminishes in strength, which can be beneficial both in the early and late stages of training. In contrast, the value network trained with discount and activation regularization fails to make unbiased predictions even at the end of the training, as shown in Figures 3 and 4.

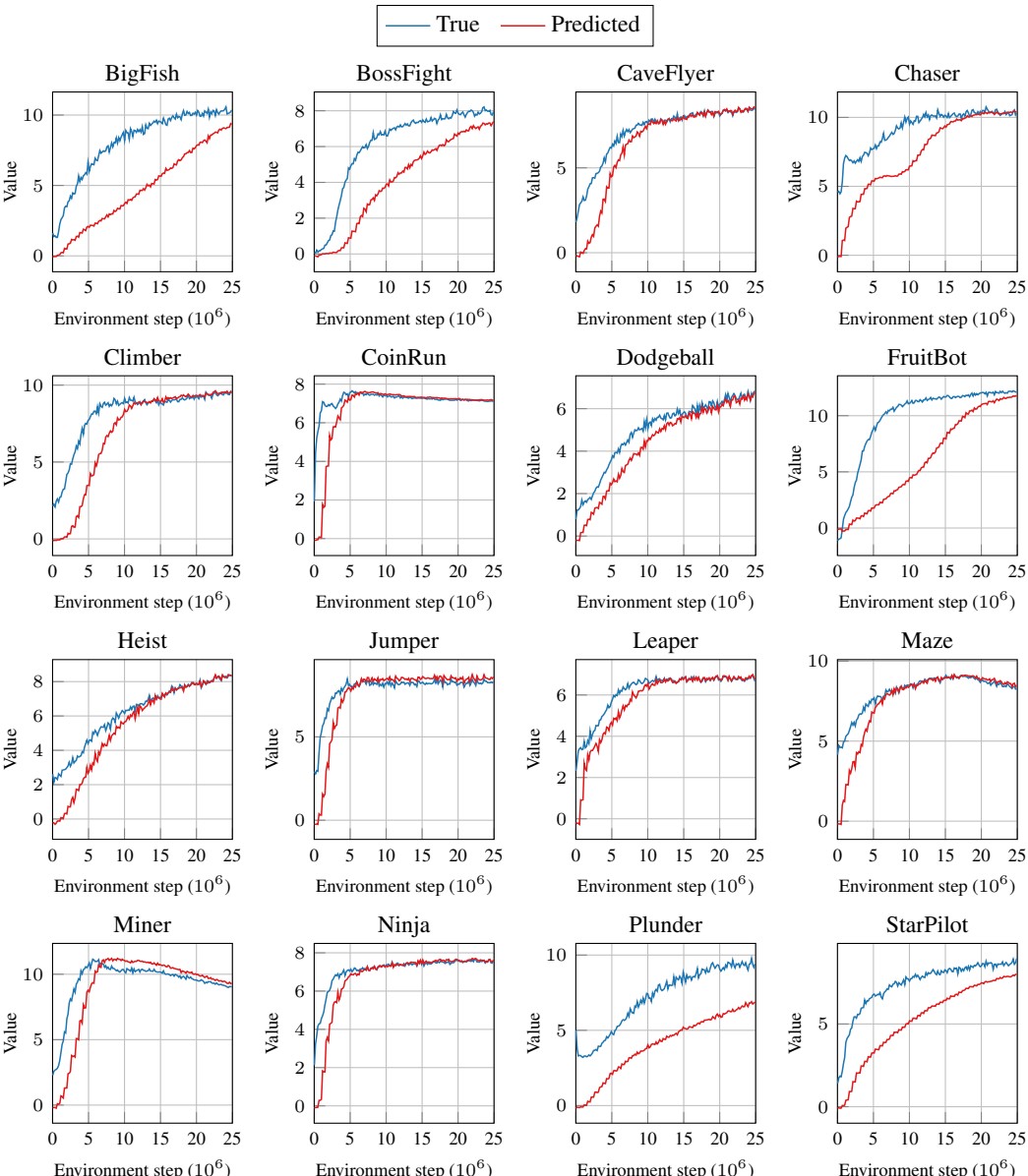

Figure 2: True and predicted values measured at the initial states of training environments for DCPG on all 16 Procgen games. The mean is computed over 10 different runs.

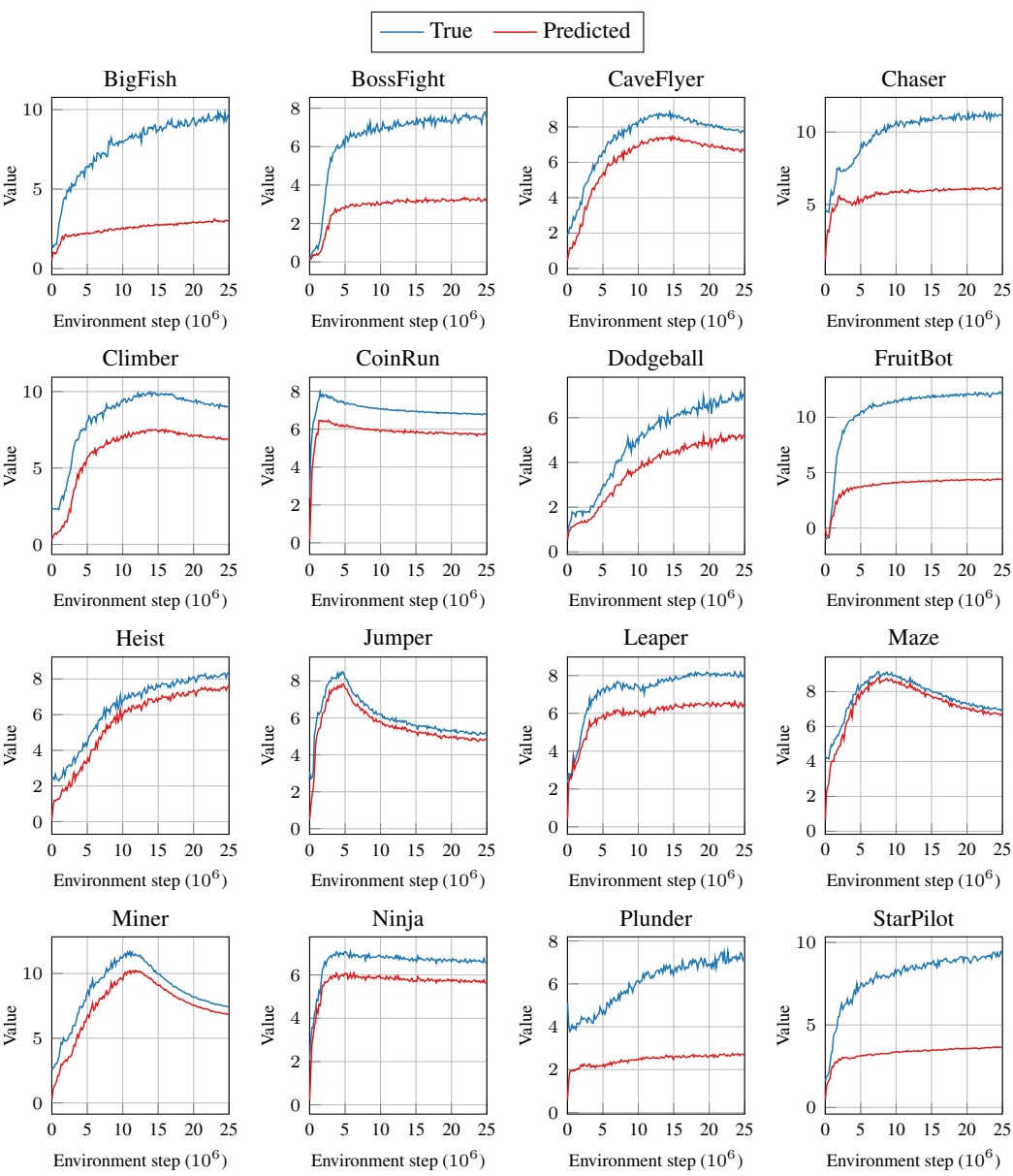

Figure 3: True and predicted values measured at the initial states of training environments for PPG with discount regularization (PPG+DR) on all 16 Procgen games. The mean is computed over 10 different runs.

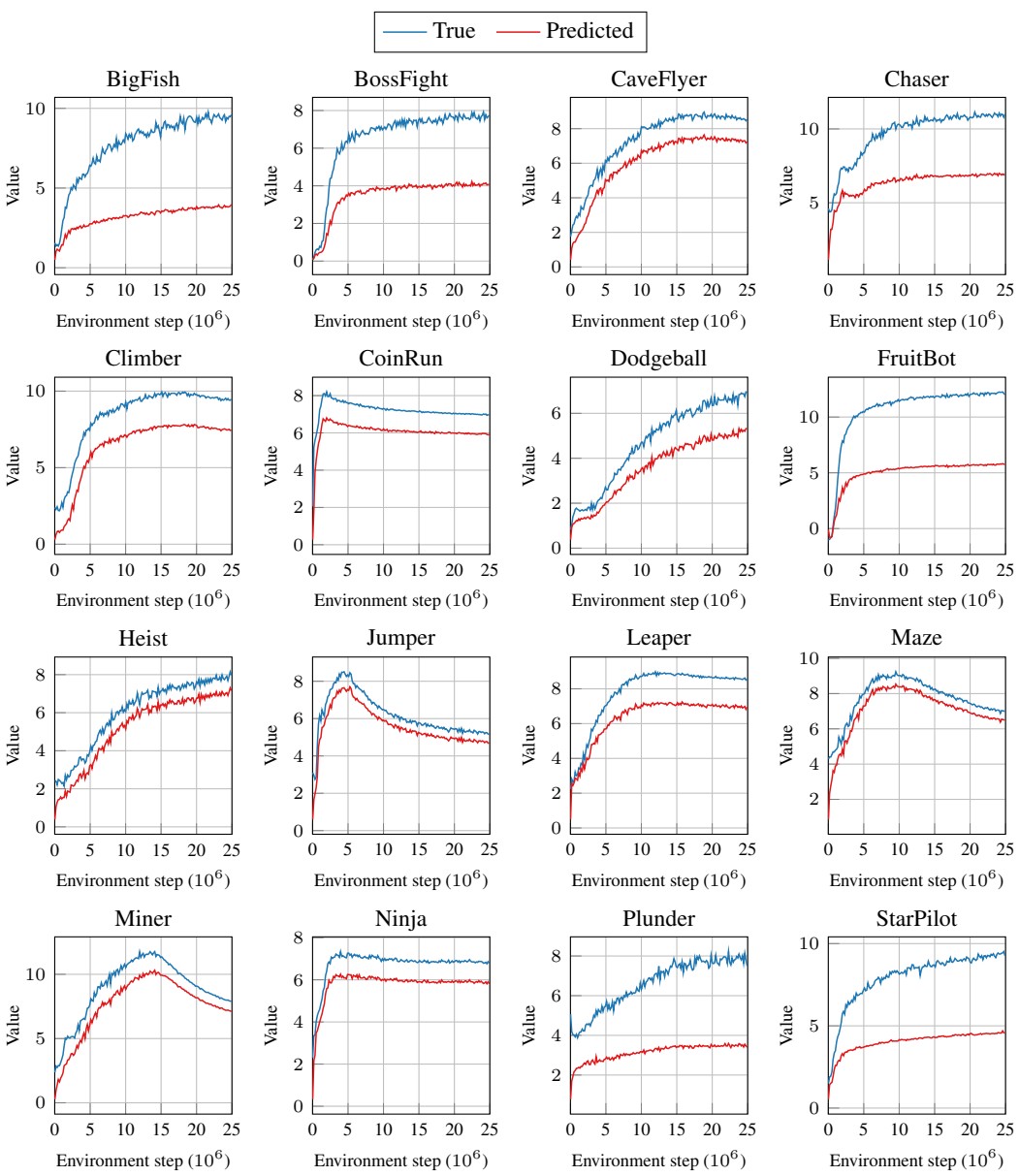

Figure 4: True and predicted values measured at the initial states of training environments for PPG with activation regularization (PPG+AR) on all 16 Procgen games. The mean is computed over 10 different runs.

# D  Implementation Details & Hyperparameters

For all experiments, we implement policy and value networks using the ResNet architecture proposed in IMPALA [5] and train the networks using Adam optimizer [6], following the standard practice in the prior works [2, 3]. We conduct all experiments using Intel Xeon Gold 5220R CPU, 64GB RAM, and NVIDIA RTX 2080 Ti GPU. We use PyTorch as a deep learning framework [8].

**PPO**  We use the implementation of PPO by Kostrikov [7]. Unless otherwise specified, we use the same hyperparameter setting provided in Cobbe et al. [2] to reproduce PPO and PPO-based methods. The values of the hyperparameters used are shown in Table 2.

Table 2: PPO hyperparameters.

| Hyperparameter | Value |
|---|---|
| Discount factor ($\gamma$) | 0.999 |
| GAE smoothing parameter ($\lambda$) | 0.95 |
| # timesteps per rollout | 256 |
| # epochs per rollout | 3 |
| # minibatches per epoch | 8 |
| Entropy bonus | 0.01 |
| PPO clip range ($\epsilon$) | 0.2 |
| Reward normalization? | Yes |
| Learning rate | 5e-4 |
| # workers | 1 |
| # environments per worker | 64 |
| Total timesteps | 25M |
| LSTM? | No |
| Frame stack? | No |

**UCB-DrAC**  We use the official implementation by the authors[1]. We use the best hyperparameter setting provided in the original paper. More specifically, we use regularization coefficient $\alpha_r = 0.1$, exploration coefficient $c = 0.1$, and sliding window size $K = 10$ for all Procgen games.

**PLR**  We reproduce the reported results of PLR using the implementation released by the authors[2]. We use the recommended hyperparameter setting presented in the original paper and use $L_1$ value loss as the scoring function, rank prioritization, temperature $\beta = 0.1$, and staleness coefficient $\rho = 0.1$ for all Procgen games.

**PPG**  We build PPG on top of the implementation of PPO. For PPO hyperparameters, we use the default hyperparameter setting in Table 2, except for the number of PPO epochs. For PPG-specific hyperparameters, we use the best hyperparameter setting provided in Cobbe et al. [3]. The values of the PPG-specific hyperparameter are shown in Table 3. The policy regularizer coefficient $\beta_\pi$ denotes the weight for the policy regularizer $C_\pi$ when jointly optimized with the auxiliary objective $J_{\text{aux}}$. We also provide the pseudocode of PPG in Algorithm 1.

**DAAC**  For DAAC and IDAAC, which adds an auxiliary regularizer to the DAAC policy network, we use the official code released by the authors[3]. We find that some hyperparameters should be set differently for each Procgen game to reproduce the results reported in the original paper. For a fair comparison with other methods, we use the same set of hyperparameters for all Procgen games with the best overall performance provided by the authors. More specifically, we use $E_\pi = 1$, $E_V = 9$, $N_\pi = 1$, $\alpha_a = 0.25$, and $\alpha_i = 0.001$. We also find that the performance of DAAC is better than IDAAC when using a single set of hyperparameters, as shown in Table 4. Therefore, we compare our methods with DAAC.

---

[1]`https://github.com/rraileanu/auto-drac`
[2]`https://github.com/facebookresearch/level-replay`
[3]`https://github.com/rraileanu/idaac`

**Algorithm 1** Phasic Policy Gradient (PPG)

**Require:** Policy network $\pi_\theta$, value network $V_\phi$, auxiliary value head $V_\theta$
1: **for** phase = $1, 2, \ldots$ **do**
2:     Initialize buffer $\mathcal{B}$
3:     **for** iter = $1, 2, \ldots, N_\pi$ **do**                                               ▷ Policy phase
4:         Sample trajectories $\tau$ using $\pi_\theta$ and compute value function target $\hat{R}_t$ for each state $s_t \in \tau$
5:         **for** epoch = $1, 2, \ldots, E_\pi$ **do**
6:             Optimize policy objective $J_\pi(\theta)$ and value objective $J_V(\phi)$ with $\tau$
7:         **end for**
8:         Add $(s_t, \hat{R}_t)$ to $\mathcal{B}$
9:     **end for**
10:    **for** iter = $1, 2, \ldots, E_{\mathrm{aux}}$ **do**                                          ▷ Auxiliary phase
11:         Optimize value objective $J_V(\phi)$, auxiliary objective $J_{\mathrm{aux}}(\theta)$, and policy regularizer $C_\pi(\theta)$ with $\mathcal{B}$
12:    **end for**
13: **end for**

Table 3: PPG-specific hyperparameters.

| Hyperparameter | Value |
|---|---|
| # policy phases per auxiliary phase ($N_\pi$) | 32 |
| # policy epochs ($E_\pi$) | 1 |
| # value epochs ($E_V$) | 1 |
| # auxiliary epochs ($E_{\mathrm{aux}}$) | 6 |
| Policy regularizer coefficient ($\beta_\pi$) | 1.0 |
| # minibatches per auxiliary epoch per $N_\pi$ | 16 |

Table 4: PPO-normalized train and test scores of DAAC and IDAAC. Each agent is trained on 200 training levels for 25M environment steps. The mean and standard deviation are computed over 10 different runs.

| | DAAC | IDAAC |
|---|---|---|
| PPO-norm train score (%) | $104.4 \pm 4.1$ | $99.7 \pm 5.4$ |
| PPO-norm test score (%) | $136.7 \pm 7.6$ | $129.9 \pm 7.8$ |

**DCPG** We build DCPG on top of the implementation of PPG. DCPG introduces one additional hyperparameter compared to PPG, named the value regularizer coefficient. The value regularizer coefficient $\beta_V$ denotes the weight for the value regularizer $C_V$ when jointly optimized with the policy objective $J_\pi$. We use the default hyperparameter setting in Table 3 for PPG hyperparameters. We set the value regularizer coefficient to $\beta_V = 1.0$ without any hyperparameter tuning.

**DDCPG** We implement the discriminator for the dynamics learning using a fully-connected layer of hidden sizes $[256, 256]$ with ReLU activation. DDCPG introduces two additional hyperparameters compared to DCPG. First, the dynamics objective coefficient $\beta_f$ denotes the weight of the dynamics objective $J_f$ when jointly optimized with the value objective $J_V$ and the policy regularizer $C_\pi$. We set the dynamics objective coefficient to $\beta_f = 1.0$ without any hyperparameter tuning. Second, the inverse dynamics coefficient $\eta$ is the hyperparameter that controls the strength of the inverse dynamics learning relative to the forward dynamics learning. We sweep the inverse dynamics coefficient within a range of $\eta \in \{0.5, 1.0\}$ and choose $\eta = 0.5$. We use the same hyperparameter setting as DCPG for the other hyperparameters.

# E  More Results on Procgen Benchmark

The training and test performance curves of DCPG, DDCPG, and baseline methods for all 16 Procgen games are shown in Figures 5 and 6, respectively. The results of UCB-DrAC and PLR are omitted in the figures for better visibility. The average training returns of DCPG, DDCPG, and all baselines are presented in Table 5. Our methods also achieve better training performance and sample efficiency than all baselines.

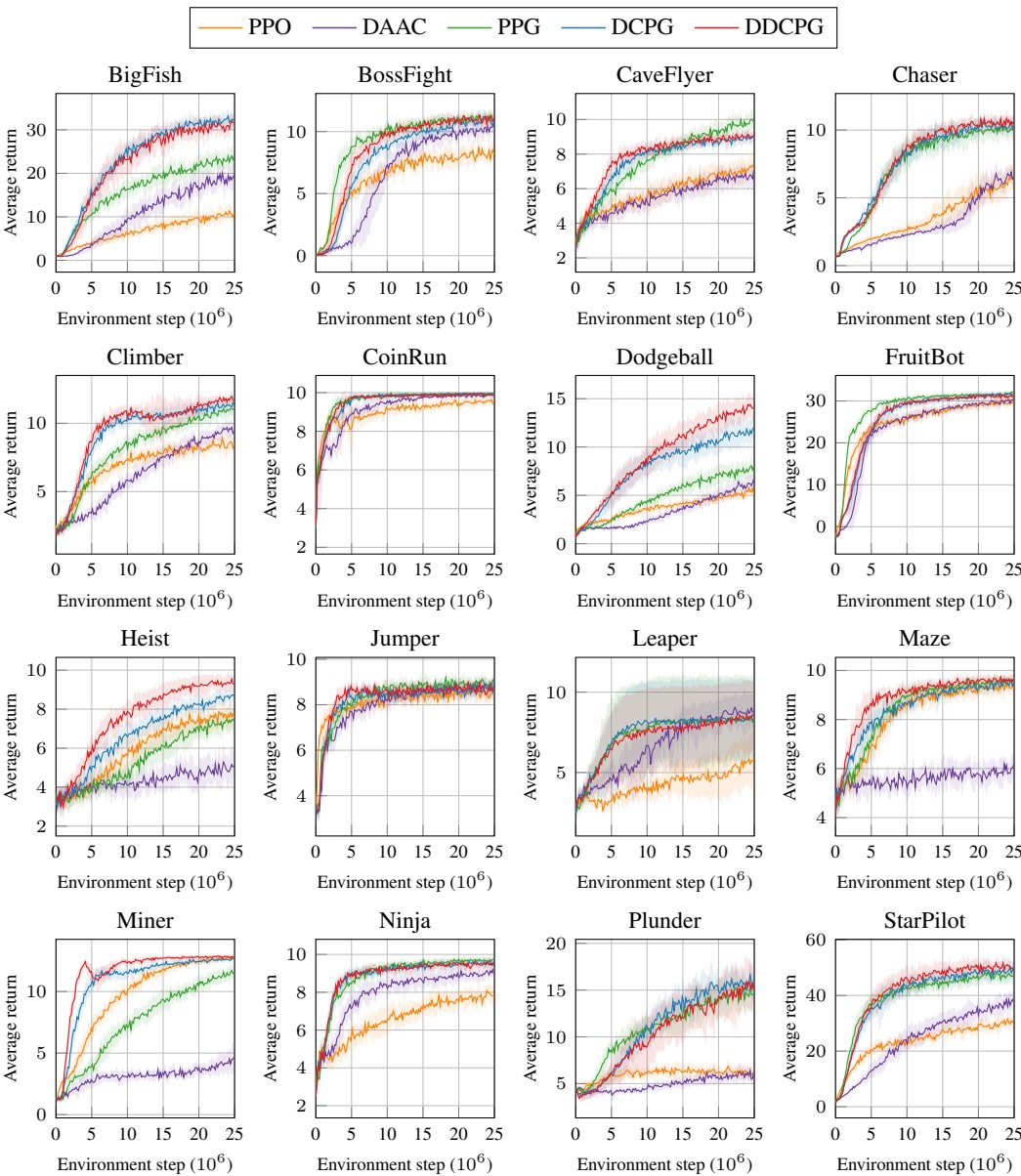

Figure 5: Training performance curves of each method on all 16 Procgen games. Each agent is trained on 200 training levels for 25M environment steps and evaluated on the same training levels. The mean and standard deviation are computed over 10 different runs.

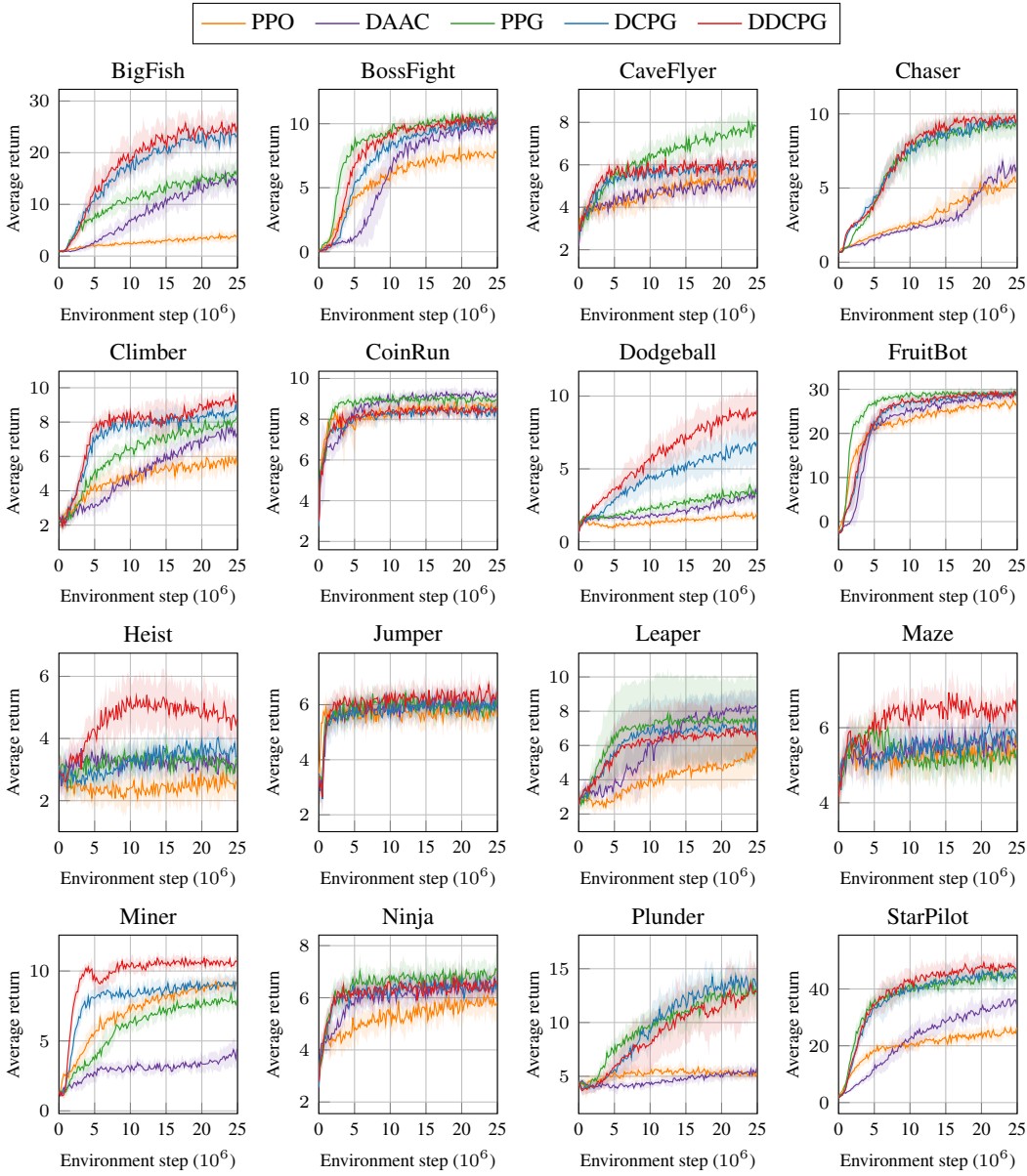

Figure 6: Test performance curves of each method on all 16 Procgen games. Each agent is trained on 200 training levels for 25M environment steps and evaluated on 100 unseen test levels. The mean and standard deviation are computed over 10 different runs.

Table 5: Average training returns of each method on all 16 Procgen games. Each agent is trained on 200 training levels for 25M environment steps and evaluated on the same training levels. The mean and standard deviation are computed over 10 different runs.

| Environment | PPO | UCB-DrAC | PLR | DAAC | PPG | DCPG | DDCPG |
|---|---|---|---|---|---|---|---|
| BigFish | $11.0 \pm 1.7$ | $12.2 \pm 3.1$ | $13.6 \pm 3.0$ | $18.4 \pm 3.3$ | $24.1 \pm 1.6$ | $32.8 \pm 2.0$ | $31.8 \pm 2.1$ |
| BossFight | $7.9 \pm 0.7$ | $8.2 \pm 0.7$ | $8.8 \pm 0.7$ | $9.9 \pm 0.7$ | $11.1 \pm 0.3$ | $10.7 \pm 0.6$ | $11.0 \pm 0.5$ |
| CaveFlyer | $7.1 \pm 0.9$ | $6.0 \pm 0.9$ | $7.3 \pm 0.5$ | $6.8 \pm 0.8$ | $10.0 \pm 0.5$ | $9.0 \pm 0.5$ | $9.0 \pm 0.3$ |
| Chaser | $6.4 \pm 0.8$ | $7.3 \pm 0.7$ | $8.0 \pm 0.6$ | $6.4 \pm 1.0$ | $10.0 \pm 0.7$ | $10.4 \pm 0.6$ | $10.6 \pm 0.5$ |
| Climber | $8.4 \pm 0.5$ | $8.4 \pm 0.6$ | $8.6 \pm 0.6$ | $9.3 \pm 0.6$ | $10.9 \pm 0.4$ | $11.3 \pm 0.5$ | $11.9 \pm 0.4$ |
| CoinRun | $9.6 \pm 0.1$ | $9.5 \pm 0.2$ | $9.4 \pm 0.3$ | $9.9 \pm 0.1$ | $10.0 \pm 0.1$ | $9.9 \pm 0.1$ | $9.9 \pm 0.1$ |
| Dodgeball | $5.5 \pm 0.7$ | $8.0 \pm 1.2$ | $5.1 \pm 0.8$ | $6.4 \pm 0.9$ | $7.9 \pm 0.7$ | $11.9 \pm 1.4$ | $14.1 \pm 1.1$ |
| FruitBot | $29.9 \pm 0.5$ | $29.1 \pm 1.3$ | $28.3 \pm 1.3$ | $29.8 \pm 1.1$ | $31.9 \pm 0.4$ | $31.9 \pm 0.4$ | $31.1 \pm 0.8$ |
| Heist | $7.6 \pm 0.6$ | $7.2 \pm 0.6$ | $8.0 \pm 0.5$ | $4.9 \pm 0.8$ | $7.4 \pm 0.6$ | $8.6 \pm 0.4$ | $9.4 \pm 0.4$ |
| Jumper | $8.7 \pm 0.3$ | $8.4 \pm 0.5$ | $8.6 \pm 0.3$ | $8.8 \pm 0.3$ | $9.1 \pm 0.3$ | $8.6 \pm 0.4$ | $8.9 \pm 0.3$ |
| Leaper | $5.8 \pm 1.5$ | $3.6 \pm 1.1$ | $7.0 \pm 0.6$ | $8.8 \pm 1.0$ | $8.2 \pm 2.9$ | $8.4 \pm 2.3$ | $8.4 \pm 2.3$ |
| Maze | $9.3 \pm 0.3$ | $8.5 \pm 0.5$ | $9.2 \pm 0.4$ | $5.7 \pm 0.4$ | $9.5 \pm 0.3$ | $9.6 \pm 0.2$ | $9.7 \pm 0.2$ |
| Miner | $12.8 \pm 0.2$ | $12.4 \pm 0.3$ | $11.4 \pm 0.6$ | $4.7 \pm 0.7$ | $11.6 \pm 0.4$ | $12.7 \pm 0.1$ | $12.8 \pm 0.1$ |
| Ninja | $7.8 \pm 0.4$ | $7.5 \pm 1.0$ | $8.2 \pm 0.4$ | $9.0 \pm 0.2$ | $9.8 \pm 0.2$ | $9.6 \pm 0.1$ | $9.6 \pm 0.2$ |
| Plunder | $6.2 \pm 0.7$ | $9.2 \pm 1.3$ | $11.0 \pm 1.1$ | $5.9 \pm 0.6$ | $14.5 \pm 2.1$ | $15.7 \pm 1.8$ | $16.0 \pm 2.3$ |
| StarPilot | $29.9 \pm 3.6$ | $30.2 \pm 2.4$ | $26.3 \pm 3.2$ | $39.0 \pm 2.5$ | $48.4 \pm 3.4$ | $50.5 \pm 1.8$ | $50.8 \pm 4.0$ |
| PPO-norm score (%) | $100.0 \pm 2.5$ | $102.6 \pm 3.4$ | $107.9 \pm 3.5$ | $104.4 \pm 4.1$ | $136.7 \pm 4.4$ | $\mathbf{148.9 \pm 4.3}$ | $\mathbf{152.7 \pm 4.4}$ |

# F  Ablation Studies

## F.1  Delayed Critic Update

To disentangle the effect of the delayed critic update on learning generalizable representations for the policy network, we introduce a variant of DCPG that employs an additional value network $V_\phi$ with a separate encoder, namely Separate DCPG. The separate value network $V_\phi$ is trained in the same way as PPG without any delayed update and used only for policy optimization. The original value network $V_\theta$ is trained with the delayed update and used only for learning representation for the policy network.

Algorithm 2 describes the detailed procedure of Separate DCPG (differences with DCPG are marked in cyan). Note that we compute two bootstrapped value function targets $\hat{R}_{t,\theta}$ and $\hat{R}_{t,\phi}$ for each state $s_t$ using the original value network $V_\theta$ and the separate value network $V_\phi$, respectively. Each value function target is stored in a buffer $\mathcal{B}$ and used to train the corresponding value network during the auxiliary phase.

---

**Algorithm 2** Separate Delayed-Critic Policy Gradient (Separate DCPG)

---

**Require:** Policy network $\pi_\theta$, value network $V_\theta$, separate value network $V_\phi$
1: **for** phase = $1, 2, \ldots$ **do**
2:     Initialize buffer $\mathcal{B}$
3:     **for** iter = $1, 2, \ldots, N_\pi$ **do**                                                         ▷ Policy phase
4:         Sample trajectories $\tau$ using $\pi_\theta$
5:         Compute value function targets $\hat{R}_{t,\theta}$ and $\hat{R}_{t,\phi}$ for each state $s_t \in \tau$
6:         **for** epoch = $1, 2, \ldots, E_\pi$ **do**
7:             Optimize policy objective $J_\pi(\theta)$ and value regularizer $C_V(\theta)$ with $\tau$
8:             Optimize value objective $J_V(\phi)$ with $\tau$
9:         **end for**
10:         Add $(s_t, \hat{R}_{t,\theta}, \hat{R}_{t,\phi})$ to $\mathcal{B}$
11:     **end for**
12:     **for** iter = $1, 2, \ldots, E_{\text{aux}}$ **do**                                                 ▷ Auxiliary phase
13:         Optimize value objective $J_V(\theta)$ and policy regularizer $C_\pi(\theta)$ with $\mathcal{B}$
14:         Optimize value objective $J_V(\phi)$ with $\mathcal{B}$
15:     **end for**
16: **end for**

---

Figures 7 and 8 demonstrate the training and test performance curves of PPG, Separate DCPG, and DCPG. We find that Separate DCPG achieves better or comparable performance in most games. This implies that the value network with delayed updates can provide better representations that generalize

well to both the training and test environments than those without delayed updates. We also provide the PPO-normalized training and test scores of PPG, Separate DCPG, and DCPG in Table 6.

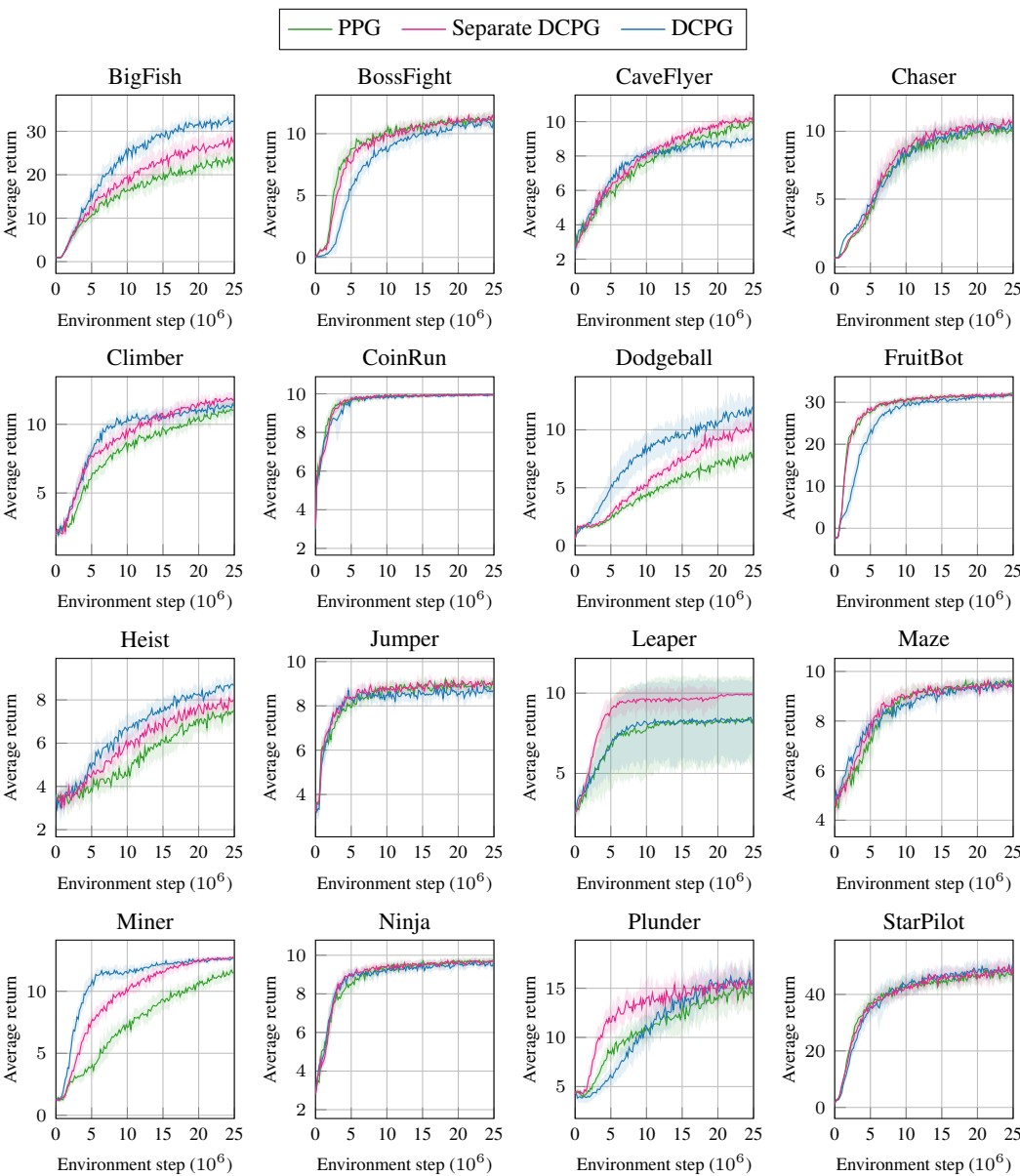

Figure 7: Training performance curves of PPG, Separate DCPG, and DCPG on all 16 procgen games. Each agent is trained on 200 training levels for 25M environment steps and evaluated on the same training levels. The mean and standard deviation are computed over 10 different runs.

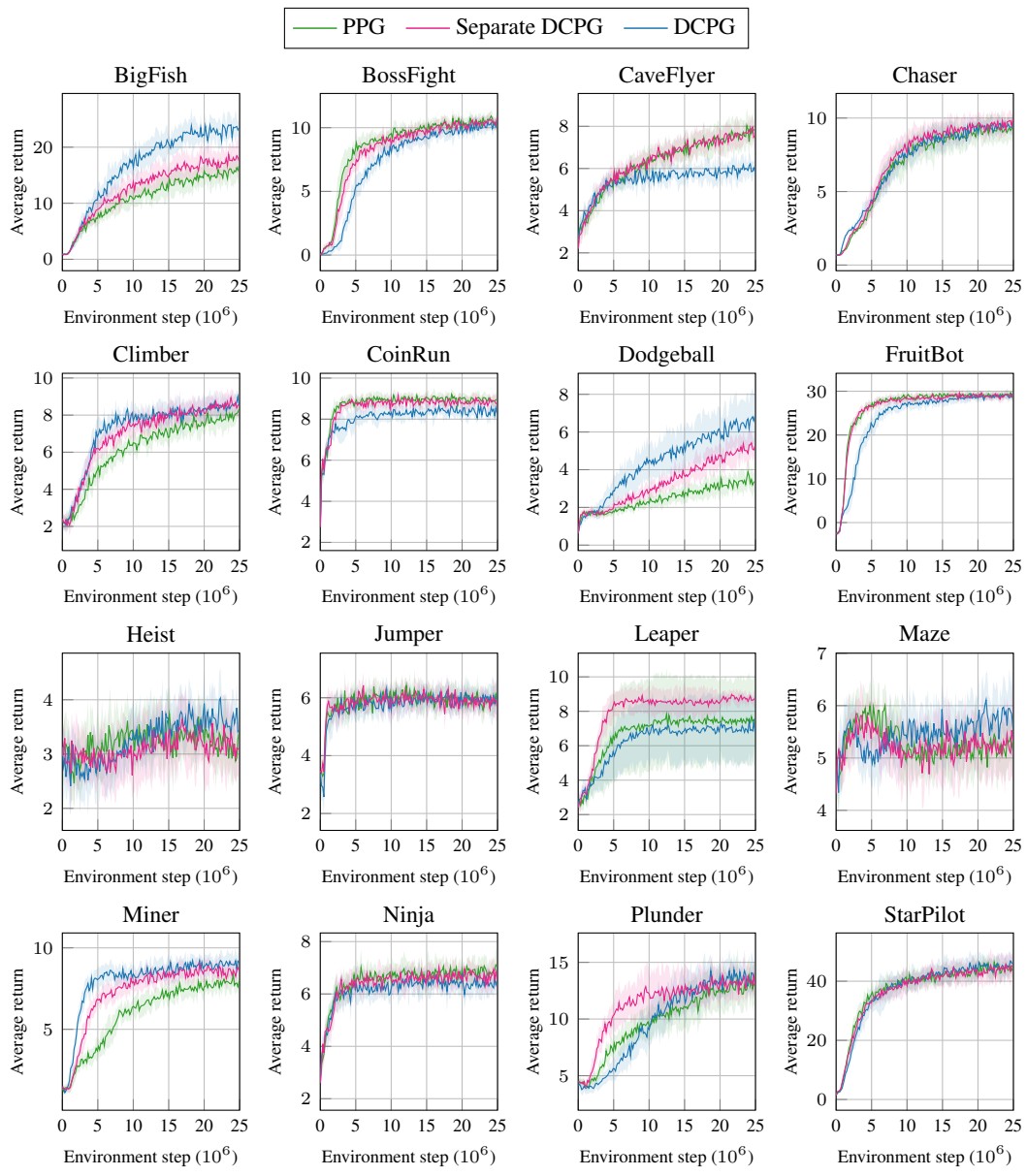

Figure 8: Test performance curves of PPG, Separate DCPG, and DCPG on all 16 procgen games. Each agent is trained on 200 training levels for 25M environment steps and evaluated on 100 unseen test levels. The mean and standard deviation are computed over 10 different runs.

Table 6: PPO-normalized training and test scores of PPG, Separate DCPG, and DCPG on all 16 Procgen games. Each agent is trained on 200 training levels for 25M environment steps. The mean and standard deviation are computed over 10 different runs.

|  | PPG | Separate DCPG | DCPG |
|---|---|---|---|
| PPO-norm train score (%) | $136.7 \pm 4.4$ | $147.5 \pm 1.9$ | $\mathbf{152.7 \pm 4.4}$ |
| PPO-norm test score (%) | $160.3 \pm 6.3$ | $174.4 \pm 4.9$ | $\mathbf{184.5 \pm 5.2}$ |

## F.2 Dynamics Learning

To demonstrate the effectiveness of learning forward and inverse dynamics with a single discriminator, we conduct experiments for DCPG with forward dynamics learning (DCPG+F), inverse dynamics learning (DCPG+I), and forward and inverse dynamics learning using two separate discriminators (DCPG+FI). The dynamics objective $J_f$ of each algorithm is defined as follows:

$$J_f(\theta) = \mathbb{E}_{s_t,a_t,s_{t+1}\sim\mathcal{B}} \left[\log\left(f_\theta\left(s_t,a_t,s_{t+1}\right)\right) + \log\left(1 - f_\theta\left(s_t,a_t,\hat{s}_{t+1}\right)\right)\right] \qquad \text{(DCPG+F)}$$

$$J_f(\theta) = \mathbb{E}_{s_t,a_t,s_{t+1}\sim\mathcal{B}} \left[\log\left(f_\theta\left(s_t,a_t,s_{t+1}\right)\right) + \log\left(1 - f_\theta\left(s_t,\hat{a}_t,s_{t+1}\right)\right)\right] \qquad \text{(DCPG+I)}$$

$$J_f(\theta) = \mathbb{E}_{s_t,a_t,s_{t+1}\sim\mathcal{B}} \left[\log\left(f_\theta\left(s_t,a_t,s_{t+1}\right)\right) + \log\left(1 - f_\theta\left(s_t,a_t,\hat{s}_{t+1}\right)\right)\right]$$
$$+ \eta\, \mathbb{E}_{s_t,a_t,s_{t+1}\sim\mathcal{B}} \left[\log\left(\tilde{f}_\theta\left(s_t,a_t,s_{t+1}\right)\right) + \log\left(1 - \tilde{f}_\theta\left(s_t,\hat{a}_t,s_{t+1}\right)\right)\right]. \qquad \text{(DCPG+FI)}$$

Note that we use an additional discriminator $\tilde{f}$ for inverse dynamics in DCPG+FI. For each algorithm, we train agents using 200 training levels for 25M environment steps on 16 Procgen games. We set the dynamics objective coefficient to $\beta_f = 1.0$ without any hyperparameter tuning. For DCPG+FI, we sweep the inverse dynamics coefficient within a range of $\eta \in \{0.5, 1.0\}$ and choose $\eta = 1.0$. Table 7 shows the PPO-normalized training and test scores of each algorithm.

Table 7: PPO-normalized training and test scores of DCPG and DCPG with dynamics learning on all 16 Procgen games. Each agent is trained on 200 training levels for 25M environment steps. The mean and standard deviation are computed over 10 different runs.

|  | DCPG | DCPG+F | DCPG+I | DCPG+FI | DDCPG |
|---|---|---|---|---|---|
| PPO-norm train score (%) | $148.9 \pm 4.3$ | $150.6 \pm 3.1$ | $149.1 \pm 3.4$ | $150.5 \pm 2.5$ | $\mathbf{152.7 \pm 4.4}$ |
| PPO-norm test score (%) | $184.5 \pm 5.2$ | $195.9 \pm 7.7$ | $185.5 \pm 6.5$ | $194.6 \pm 6.2$ | $\mathbf{202.2 \pm 10.2}$ |

Our intuition of jointly learning forward and inverse dynamics using a single discriminator is that naively learning the forward dynamics will discard action information and capture only the proximity of two consecutive states in the latent space, not the dynamics. Also, additional training of the inverse dynamics with a separate discriminator cannot completely resolve this problem.

To validate this, we train three types of DCPG agents with dynamics learning, DCPG+F, DCPG+FI, and DDCPG on BigFish and count the number of OOD actions that the forward dynamics discriminator determines to be valid given a transition $(s_t, a_t, s_{t+1})$ from the training environments, *i.e.*, $\sum_{\hat{a}_t \neq a_t} \mathbb{1}[f(s_t, \hat{a}_t, s_{t+1}) > 0.5]$. Note that there are 9 different actions (8 directional moves and 1 do nothing) in BigFish. Table 8 shows that the discriminators of DCPG+F and DCPG+FI determine multiple OOD actions to be valid, implying that it does not fully utilize the action information.

Table 8: The number of OOD actions classified as valid for DCPG+F, DCPG+FI, and DDCPG on BigFish. Each agent is trained on 200 training levels for 25M environment steps.

|  | DCPG+F | DCPG+FI | DDCPG |
|---|---|---|---|
| # OOD actions ($\downarrow$) | $7.05 \pm 1.47$ | $2.49 \pm 1.38$ | $\mathbf{0.74 \pm 0.75}$ |

Furthermore, we train PPG agents with our proposed dynamics learning (named DPPG) to check whether the effect of the delayed critic update and the dynamics learning are complementary. Table 9 shows that dynamics learning is also helpful to PPG, while the extent of performance improvement is smaller than DCPG. It implies that the effects of the delayed value update and the dynamics learning are synergistic.

Table 9: PPO-normalized test scores of PPG, DPPG, DCPG, and DDCPG on all 16 Procgen games. Each agent is trained on 200 training levels for 25M environment steps. The mean and standard deviation are computed over 10 different runs.

|  | PPG | DPPG | DCPG | DDCPG |
|---|---|---|---|---|
| PPO-norm score (%) | $160.3 \pm 6.3$ | $171.7 \pm 4.9$ | $184.5 \pm 5.2$ | $202.2 \pm 10.2$ |

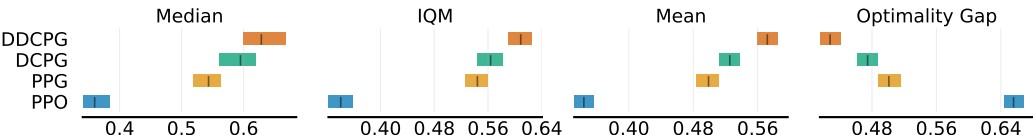

Figure 9: Median/IQM/Mean of Min-Max normalized scores with 95% confidence intervals for PPO, PPG, DCPG, and DDCPG on all 16 Procgen games. Each statistic is computed over 10 seeds.

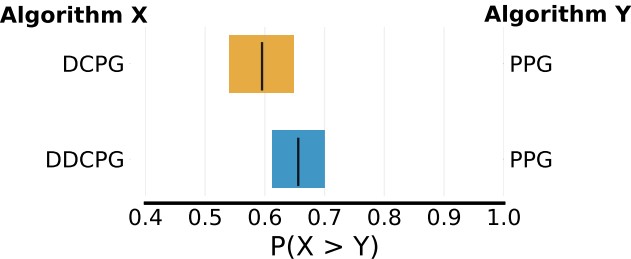

Figure 10: Probability of improvement of DCPG and DDCPG compared to PPG on all 16 Procgen games. Each statistic is computed over 10 seeds.

## G   Evaluation using RLiable [1]

We also evaluated our experimental results by normalizing the average returns based on the possible minimum and maximum scores for each game and analyzing the min-max normalized scores using the RLiable library[4]. Figure 9 reports the Median, IQM, and Mean scores of PPO, PPG, and our methods, showing that the performance improvements of our methods are statistically significant in terms of all evaluation metrics. Figure 10 describes the probability of improvement plots comparing our methods to PPG, showing that our methods are likely to improve upon PPG.