# OpenReview forum: "Rethinking Value Function Learning for Generalization in Reinforcement Learning"
_NeurIPS.cc/2022/Conference — NeurIPS 2022 Accept_

### Official Review · Reviewer_M6vK · 2022-07-01

**Rating:** 7
**Confidence:** 4
**Soundness:** 3 good
**Presentation:** 4 excellent
**Contribution:** 3 good

**Summary:**

In this paper, it is first shown that the value function is easier to overfit compared to the policy function.
Based on this observation, a new policy gradient algorithm called Delayed-Critic Policy Gradient (DCPG) is proposed, which reduces the overfitting of the value function by less frequent updates with more training data.
Moreover, a single discriminator is applied to learn both forward and inverse dynamics of environments, which helps to generate better representations and improve the performance of DCPG across different tasks.

**Questions:**

- In Figure 4, how are the true values computed? How to make sure that these computed values are true values during training?
- Do DCPG and DDCPG use a larger replay buffer than the baseline algorithms, such as PPO and PPG?

**Limitations:**

In Table 2, the error bars of some results overlap with each, making the claim in Section 5.4 less convincing.

**Strengths And Weaknesses:**

This work presents two novel policy gradient algorithms --- DCPG and DDCPG.
Compared to PPG, DCPG has a more compact model structure with only one encoder, while achieving comparable or better performance on different tasks.
Moreover, by combining the discriminator that learns environmental dynamics, DDCPG boosts the performance of DCPG furthermore.
This paper is clearly written with a good flow.
The analysis experiments and the ablation studies are also helpful in understanding the effectiveness of DCPG and DDCPG.

There are several minor issues.
In Section 2.2, the clip operation is not explained though it might be obvious to readers familiar with PPO.
In Section 4.3, both the encoder $e_\theta(\cdot)$ and the dynamics head $f_\theta(\cdot, \cdot, \cdot)$ are parameterized by $\theta$. Is the encoder the same one as shown in Figure 1(c)? If so, how could the same $\theta$ be used to parameterize a different function, i.e. $f_\theta(\cdot, \cdot, \cdot)$? It is better to present the model structure of DDCPG as well to clarify.

---

> ### Author Response · Authors · 2022-08-02
> **Response to Reviewer M6vK**
>
> We thank the reviewer for the helpful and constructive feedback. We are encouraged by the reviewer’s positive comments that our proposed methods are novel and our paper is clearly written with a good flow. We would like to address the questions raised by the reviewer below.
>
> ---
> **1. The clip operation in PPO is not explained.**
>
> Thank you for the suggestion. We will add an explanation in Section 2.2 that the clip operation is designed to update the policy conservatively.
>
> **2. Both the encoder and the dynamics head are parameterized by $\theta$.**
>
> Thank you for the recommendation. The encoder $e_\theta(\cdot)$ is the same one as shown in Figure 1(c). More specifically, the network for DDCPG is comprised of a single convolutional encoder and 3 MLP heads for policy, value, and dynamics. We have denoted all parameters of the encoder and the heads by $\theta$ for the sake of notational simplicity, but we agree that this expression can be misleading. We will also present the network architecture of DDCPG in FIgure 1 and use different notation for parameters of the encoder and the MLP heads in the revised version.
>
> **3. How are the true values computed?**
>
> As noted in Supplementary C, we estimate the true values of the on-policy value function for the initial state by collecting trajectories using the current policy and computing the average discounted returns.
>
> **4. Do DCPG and DDCPG use a larger replay buffer than the baseline algorithms?**
>
> We first clarify that PPO trains a policy using only on-policy data and does not have any replay buffer. For DCPG and DDCPG, we use the same size of replay buffer as PPG for a fair comparison.
>
> **5. In Table 2, the error bars of some results overlap with each.**
>
> We find that the score of DDCPG is statistically greater than other baselines. We conduct an unpaired t-test to compare the mean scores of DDCPG and the second-best DCPG+F, increasing the number of training seeds from 10 to 20 to obtain a reasonable number of samples for the test. The table below shows that the difference between the mean scores of DDCPG and DCPG+F is statistically significant.
>
> | DCPG+F score | DDCPG score | P-value (two-tailed)|
> |---|---|---|
> | 191.2 ± 7.4 | 197.4 ± 8.0 | 0.015 |
>
> ---
> We thank the reviewer again for the time and effort spent on providing valuable feedback. We hope that our answers address all the reviewer's points.
>
> Paper7811 Authors

---

### Official Review · Reviewer_SbES · 2022-07-07

**Rating:** 5
**Confidence:** 4
**Soundness:** 3 good
**Presentation:** 3 good
**Contribution:** 2 fair

**Summary:**

This paper proposes a new approach for RL in procedurally generated environments with empirical evidence. At its core is the problem of the critic learning problem and potential overfitting. The proposed approach mainly builds on the Phasic Policy Gradient method by unifying it. The method is compared against natural baselines and other environment-relevant methods. The experiment section also shows several ablation studies.

**Questions:**

The above comment does not detract from the pertinence of the method, but I have a few comprehension questions or some comments about the paper, please see below:

1. Some relevant works tackling the accuracy of the critic and its impact on the agent's learning seem to be missing: [1] with an empirical analysis of the phenomena and [2] with the introduction of a critic trained only using the residual variance to speed up learning.
2. Could the authors elaborate on the motivation for adding the Forward/Inverse Dynamics Learning task? The paper's focus is mainly on value function overfitting and its regularization - if any, could the author help understand to what extent does using this auxiliary task help avoid overfitting?
3. Have the authors considered applying this Forward/Inverse Dynamics Learning task to PPG? It would be interesting to know if using this component also helps PPG and to what extent (maybe even more than DCPG).
4. Table 2 is quite interesting. Could the authors provide an intuition about why learning the inverse dynamics separately seems to be worse than learning it jointly with the forward dynamics with a single discriminator? Is the total number of parameters for the “dynamics learning module” kept the same? i.e. how many parameters does DCPG+FI have vs. DDCPG?


[1] Ilyas, A., Engstrom, L., Santurkar, S., Tsipras, D., Janoos, F., Rudolph, L., & Madry, A. (2020). A closer look at deep policy gradients. In International Conference on Learning Representations.
[2] Flet-Berliac, Y., Ouhamma, R., Maillard, O.-M., & Preux, P. (2021). Learning value functions in deep policy gradients using residual variance. In International Conference on Learning Representations.

**Limitations:**

The potential negative societal impact or the limitations of the work have not been addressed.

**Strengths And Weaknesses:**

The contribution of this paper is original and appears relevant to the community. It nicely builds on a battery of previous work, which has some overfitting limitations. The empirical analysis is quite furnished and gives very some interesting insights. However, the improvements provided by DCPG or DDCPG over the baselines appear to be less important than what is claimed by the authors. In particular, the statistical significance in Tables 1, 2, and 5 depicted by bold text and the empirical analysis in Section 5.2 cannot be conclusive. Statistically speaking when compared to the baselines, DCPG appears to improve on 2/18 tasks and DDCPG on 4/18. Indeed, unfortunately, the stds appear in many instances to be generally too significant to draw any statistical conclusion. We develop more points below. Regarding the quality, overall, the paper is also mostly well written and develops its ideas clearly.

---

> ### Author Response · Authors · 2022-08-02
> **Response to Reviewer SbES**
>
> We thank the reviewer for the valuable and insightful feedback. We appreciate the encouraging comments (“the contribution of this paper is original”, “the empirical analysis is quite furnished and gives very some interesting insights”). We would like to address the questions and concerns the reviewer has below.
>
> ---
> **1. Some relevant works tackling the accuracy of the critic and impact on the agent’s learning seem to be missing.**
>
> Thank you so much for the advice. In particular, the value function analysis of [1] in the single-environment setting is closely related to our work and corroborates our claim in the multi-environment setting. We will clearly state these related works in the first paragraph of Section 3.
>
> **2. Motivation for adding the dynamics learning task.**
>
> In the context of multi-task learning, it is well-known that learning an auxiliary task using shared parameters with the main task serves as a regularizer that prevents the overfitting in the main task [2]. Motivated by this, we consider jointly learning the value function (main) and the forward/inverse dynamics (auxiliary) using the same convolutional encoder. We will explain this motivation more clearly in Section 4.3.
>
> **3. Have the authors considered applying the dynamics learning task to PPG?**
>
> Thank you for the suggestion. We train PPG agents with the forward/inverse dynamics learning for all Procgen games and reported their performance in the table below. It shows that the dynamics learning is also helpful to PPG, while the extent of performance improvement is smaller than DCPG. It also implies that the effects of the delayed value update and the dynamics learning are synergistic.
>
> | | PPG | PPG+Dynamics | DCPG | DCPG+Dynamics (DDCPG)
> |---|---|---|---|---|
> | Score ($\uparrow$) | 160.3 ± 6.3 | 171.7 ± 4.9 | 184.5 ± 5.2 | 202.2 ± 10.2 |
>
> **4. Could the author provide an intuition about why learning the inverse dynamics separately seems to be worse?**
>
> We first clarify that each discriminator has the same number of parameters. So, DCPG+FI has twice as many parameters as DDCPG. Our intuition is that naively learning the forward dynamics will discard the information about the action and captures only the proximity of two consecutive states, not the dynamics. Also, additional training of the inverse dynamics with a separate discriminator cannot completely resolve this problem.
>
> To validate this, we train three types of DCPG agents with dynamics learning, DCPG+F, DCPG+FI, and DDCPG on the BigFish game and measure the number of OOD actions that the forward dynamics discriminator determines to be valid given a transition $(s, a, s’)$, i.e., $ \sum_{\hat{a} \neq a} \mathbb{1}[D(s, \hat{a}, s’) > 0.5]$, where $D$ is the discriminator. Note that there are 9 different actions (8 directional moves and 1 do nothing) in the BigFish. The table below shows that the forward dynamics discriminator of DCPG+F or DCPG+FI determines multiple OOD actions to be valid, which implies that it does not fully utilize the action information and fails to learn the dynamics. On the other hand, DDCPG does not discard the action information and successfully distinguishes OOD actions as invalid. We will add this analysis in the revised version.
>
> | | DCPG+F | DCPG+FI | DDCPG |
> | --- | --- | --- | --- |
>  \# OOD actions ($\downarrow$) | 7.05 ± 1.47 | 2.49 ± 1.38 | 0.74 ± 0.75 |
>
> **5. The stds appear to be generally too significant to draw any statistical conclusion.**
>
> We acknowledge that the scores for each individual game have high standard deviations. In the revised version, we consider not highlighting individual game scores in Table 1 and rewrite Section 5.2 to avoid misleading.
>
> ---
> Thank you again for your constructive comments, which help us to improve our paper’s quality. We hope that our answers address all the reviewer's points.
>
> Paper7811 Authors
>
> ---
> References
>
> [1] Ilyas, et al., A Closer Look at Deep Policy Gradients, ICLR 2020.
> [2] Ruder, An Overview of Multi-Task Learning in Deep Neural Networks, 2017.

---

> > ### Comment · Reviewer_SbES · 2022-08-07
> > **Thank you to the authors and hereby acknowledging the updated version**
> >
> > I want to thank the authors for their response and the time spent on the revision.
> >
> > I have updated my score accordingly.

---

### Official Review · Reviewer_FX6f · 2022-07-21

**Rating:** 6
**Confidence:** 4
**Soundness:** 3 good
**Presentation:** 3 good
**Contribution:** 2 fair

**Summary:**

This paper claims that training policy and value networks separately for actor critic algorithms like PPO or PPG can lead to overfitting of the value network in procedural generalization tasks like Procgen. To get around the overfitting issue, it proposes to update the value network less often and with more data. It also proposes to add a self-supervised objective to further improve generalization performance, and observes modest gains on some games in Procgen.

**Questions:**

* The state-of-the-art claim might not stack well against MuZero++ (https://arxiv.org/abs/2111.01587) given the gap b/w MuZero++ and PPG/PLR/etc. However, it looks like the authors were not simply aware of this work, and DDCPG being model-free is not directly comparable to MuZero++ anyways, so I don't hold any negative points for this one if its mentioned in the revisions.
* It would be nice to see if and how does the performance changes as the model size and data size is increased. Perhaps you can try increasing the IMPALA network size systematically or using a ResNet, and similarly train on more amounts of data.
* Generalized in RL is an overloaded term, and this paper studies a very specific form of generalization (i.e. observational or procedural generalization). There are other forms of generalization such as changing the task or reward functions. Make sure that this distinction is clear.
* The self-supervised objective should be appropriately positioned with respect to prior work listed above.
* It would be a good idea to use the rliable colab (https://arxiv.org/abs/2108.13264) to plot the results since it has been found that standard reporting of results can be misleading. Specifically, please include the IQM and probability of improvement plots. Also, following the recommendation its better to report Min-max rather than PPO normalized scores.
* The bolding of scores in Table 1 relies on mean scores it looks like, which can be very misleading. If two methods have overlapping mean and std devs, they should both be bolded or neither since variance can be high for these scores.

**Ethics Review Area:**

["I don’t know"]

**Strengths And Weaknesses:**

Strengths:
* The paper makes an interesting empirical observation of overfitting in separate update PPG agents that could harm their generalization performance.
* The paper is very clearly written except for section 4.3 and gets the idea across clearly. Figure 1 is a great summary of the proposed modification with clear differences to PPG and PPO.
* The fix for the overfitting issue is straightforward and easy to implement.
* The experiments are performed on a well benchmarked environment suite (Procgen) and include informative ablations. There are also a bunch of valuable auxiliary experiments investigation value function learning.

Weaknesses:
* The contribution of delayed value updates itself seems not quite significant, very specific to PPG and its unclear how well it would generalize to other algorithms.
* The overfitting issue might just vanish as we scale the model and data size for these agents. So I fear the delayed value update contribution might end up being a "regularization game" for this specific PPG setup and end up getting "bitter lesson'ed" in the long run.
* Note: The above two are subjective opinions on significance of the contributions.
* The improvement for DCPG over PPG seems marginal, especially when we look at individual performance curves in Figure 6 of the appendix. It seems like there are only 3-4 out of 16 games where the regularization has a clear and significant benefit.
* The contribution a self-supervised objective with learned dynamics has been explored multiple times in the literature (SPR (https://arxiv.org/abs/2007.05929), PBL (https://arxiv.org/abs/2004.14646), DeepMDP (https://arxiv.org/abs/1906.02736) etc. Moreover, it has been explored explicitly in Procgen as well in MuZero++ (https://arxiv.org/abs/2111.01587) and DRIML (https://arxiv.org/abs/2006.07217). The idea of using a discriminator for this task was proposed in CPC|Action, and evaluated in Procgen in DRIML and ablated in MuZero++. Using an inverse modelling objective along with the forward objective was also proposed in SGI (https://arxiv.org/abs/2106.04799). The paper unfortunately doesn't mention or appropriately position itself against these works.
* The reporting of results does not follow the recently proposed recommended practices as described in rliable (https://github.com/google-research/rliable), and thus might be prone to making incorrect conclusions. More details below.

---

> ### Author Response · Authors · 2022-08-02
> **Response to Reviewer FX6f**
>
> We thank the reviewer for the valuable and constructive feedback. We appreciate the encouraging comments (“the paper makes an interesting empirical observation”, “the paper is very clearly written”, “the fix is straightforward and easy to implement”). We would like to address the reviewer’s concerns below.
>
> ---
> **1. The state-of-the-art claim might not stack well against MuZero++.**
>
> Thank you for pointing this out. We will clearly state in Section 1 that our work mainly focuses on model-free RL algorithms and mention in Section 6 that MuZero++ has a better performance compared to model-free algorithms such as PPO or PLR, while MuZero++ requires a larger network architecture.
>
> **2. It would be nice to see if and how the performance changes as the model size or data size is increased.**
>
> Thank you for the recommendation. We conduct experiments on larger network architecture by increasing the width of the IMPALA network from [16, 32, 32] to [32, 64, 64] (denoted by 2x) and report the PPO normalized and min-max normalized scores in the table below. The results show that while increasing the number of parameters improves the performance of the baseline PPG, the performance gap between DCPG and PPG remains constant. This implies that the effect of increasing network size and the delayed critic update are orthogonal.
>
> | | PPG | PPG (2x) | DCPG | DCPG (2x) |
> |---|---|---|---|---|
> PPO normalized score | 160.3 ± 6.3 | 179.5 ± 6.8 | 184.5 ± 5.2 | 203.1 ± 8.5 |
> Min-max normalized score | 0.499 ± 0.025 | 0.553 ± 0.023 | 0.525 ± 0.024 | 0.574 ± 0.018 |
>
> For data size, if the reviewer meant increasing the batch size of the baseline method, we find that it is not helpful to improve the performance. We train PPG agents with a larger batch size, increasing the batch size for the critic in the policy phase to the same level as DCPG (16,384 -> 524,288) and find that the PPO-normalized score decreases from 160.3 to 137.1. If the reviewer meant increasing the number of interactions with environments, please let us know and we will do experiments as soon as possible.
>
> **3. Generalized in RL is an overloaded term.**
>
> Thank you for the suggestion. We will clarify in Section 2.1 that our work tackles the observation or state space variation, not dynamics or reward function variation, and consider using the term “procedural generalization” proposed in [1] throughout the paper.
>
> **4. The self-supervised objective should be appropriately positioned with respect to prior work.**
>
> Our claim on the self-supervised objective is that
> - The forward dynamics objective using a discriminator is likely to learn the proximity between two consecutive states, not the dynamics.
> - Jointly optimizing the forward and inverse dynamics objective using a single discriminator enables the discriminator to learn the dynamic and improves the procedural generalization performance of RL agents.
>
> We will rewrite the second paragraph in Section 6 and explain our claim more clearly in the revised version. We also cite the relevant works on dynamics learning appropriately.
>
> **5. It would be a good idea to use the RLiable.**
>
> Following your recommendation, we re-evaluated our experiments by normalizing the average returns based on the possible minimum and maximum scores for each game and analyzing the min-max normalized scores using the RLiable library (https://github.com/google-research/rliable). The results can be found in the following links.
>
> Median/IQM/Mean scores: https://drive.google.com/file/d/1Y-GoeWnfcYYEd9grT4YsisnWf5GzdYzg/view?usp=sharing
> Probability of improvement: https://drive.google.com/file/d/1Uxq2xOraBc-cU84PO-UdrvDjAaw1W_f1/view?usp=sharing
>
> The first figure shows that the performance gap between each of our methods and PPG is statistically significant in terms of all evaluation metrics. The second figure shows that our methods are likely to improve upon PPG. We will report the min-max normalized score as well as the PPO normalized score and include the Median/IQM/Mean and probability of improvement plots in the revised version.
>
> **The Bolding of scores in Table 1 relies on mean scores, which can be very misleading.**
>
> We will highlight a method in boldface if the mean plus one std of the method exceeds the mean of the best method in the revised version. Alternatively, we consider not highlighting individual game scores but highlighting only the overall scores averaged over all games, following the practice in [1].
>
> ---
> We again thank the reviewer for providing helpful suggestions, which truly improve the quality of our paper. We hope that our answers address all the reviewer's points.
>
> Paper7811 Authors
>
> ---
> References
>
> [1] Anand, et al., Procedural Generalization by Planning with Self-Supervised World Models, ICLR 2022.

---

> > ### Comment · Reviewer_FX6f · 2022-08-07
> > **Thanks for the clarifications**
> >
> > Thanks for the clarifications, I indeed meant more environment interactions not a bigger batch size.
> >
> > >The forward dynamics objective using a discriminator is likely to learn the proximity between two consecutive states, not the dynamics.
> >
> > If this is the contribution, then again I would like to emphasize that it has been explored before in the literature. See Time Contrastive Networks, CPC, or ST-DIM. It is fine to use existing losses in conjunction along with your method without claiming a new ground of novelty.
> >
> > The rest of the responses answer my questions well, I will wait for the final round of revisions before revising my score.

---

> > > ### Author Response · Authors · 2022-08-09
> > > **Response to Reviewer FX6f**
> > >
> > > We thank the reviewer for taking additional time to read our response and willing to further discuss with us. We would like to address the reviewer's remaining questions below.
> > >
> > > ---
> > > **1. Increasing the number of environment interactions**
> > >
> > > We train PPG and DCPG agents by increasing the number of environment interactions from 25M to 50M and report the PPO normalized and Min-Max normalized scores. As shown in the table below, DCPG achieves better performance than PPG in terms of PPO normalized score while the performance gap decreases as the number of environment interactions increases. Also, as shown in the attached link, DCPG has better median performance than PPG but comparable mean and IQM performances in terms of Min-Max normalized score. Still, our method can acquire the same level of performance with a fewer number of environment interactions. We agree with the reviewer’s opinion that the data size can easily scale with increased computational power in a simulated environment. However, as mentioned in [1], collecting data in a real-world environment can be expensive (requires human labor) or dangerous. We would like to emphasize that developing a more sample-efficient algorithm that can achieve good generalization performance is also important.
> > >
> > > |   | PPG (25M)  | DCPG (25M)  | PPG (50M)  | DCPG (50M)  |
> > > |---|---|---|---|---|
> > > | PPO normalized score  | 160.3 ± 6.3 | 184.5 ± 5.2 | 179.2 ± 4.8  | 198.8 ± 8.2 |
> > >
> > > Min-Max normalized scores: https://drive.google.com/file/d/13blajHfhaX7WwnkEwvdni4dKg1tNIUo-/view?usp=sharing
> > >
> > > **2. About dynamics learning**
> > >
> > > We first clarify that our objective is to train a discriminator that predicts future states by utilizing both the information of the current state and the action taken, which is different from CPC [2] or ST-DIM [3], whose objective is to predict future states using only the information of the current state. Also, our dynamics learning differs from CPC|Action [4] in that our method requires sampling OOD actions. More specifically, we found that the forward dynamics discriminator without the OOD action sampling tends to discard the information about the action and only captures the proximity of two consecutive states, not the dynamics. We compare two types of dynamics learning, DCPG+F (without OOD action sampling) and DDCPG (with OOD action sampling) on BigFish and measure the number of OOD actions that the forward dynamics discriminator determines to be valid given a transition (s, a, s’), i.e., $\sum_{\hat{a} \neq a} \mathbb{1}[f(s, \hat{a}, s') > 0.5]$, where $f$ is the discriminator. Note that there are 9 different actions (8 directional moves and 1 do nothing) in BigFish. The table below shows that the discriminator of DCPG+F determines most of the OOD actions (7 out of 8) to be valid, indicating that it does not fully utilize the action information and fails to learn the dynamics. On the other hand, DDCPG does not discard the action information and successfully distinguishes OOD actions as invalid. We empirically found that this OOD action sampling is beneficial for improving observational generalization performance.
> > >
> > > |   | DCPG+F | DDCPG |
> > > |---|---|---|
> > > | \# OOD actions ($\downarrow$) | 7.05 ± 1.47 | 0.74 ± 0.75 |
> > >
> > > ---
> > >
> > > Again, we really appreciate the reviewer's insightful questions, which greatly help us to improve our paper. We hope that our response above has addressed all of the remaining concerns.
> > >
> > > Paper7811 Authors
> > >
> > > ---
> > > References
> > >
> > > [1] Levine, et al., ​​Offline Reinforcement Learning: Tutorial, Review, and Perspectives on Open Problems, 2020 \
> > > [2] van den Oord, et al., Representation Learning with Contrastive Predictive Coding, 2018 \
> > > [3] Anand, et al., Unsupervised State Representation Learning in Atari, NeurIPS 2019 \
> > > [4] Guo, et al., Neural Predictive Belief Representations, 2018

---

> > > > ### Comment · Reviewer_FX6f · 2022-08-09
> > > > **Response**
> > > >
> > > > Thanks for the additional experiments with more data. In the plots for min-max normalized scores, it looks like DCPG doesn't outperform PPG on IQM and Mean scores. This is an important point that should be highlighted in the main text. I recommend including the plots in the appendix as well.
> > > >
> > > > As for the differences with CPC | Action, I think you could be more explicit in the text about it. Perhaps directly clarifying this in sec 4.3 and not relegating to the related work section would be ideal.
> > > >
> > > > Overall, the manuscript has improved, the authors have added better evaluation metrics and performed additional experiments around some questions. I am still skeptical of the overall idea, it still feels like a regularization hack to me, and experiments with more data do seem to indicate that as well. But the experiments are well executed, the paper is well written, and we have some good evidence of DCPG helping in certain regimes. Taking that into account, I am improving my score to a 6 now.

---

### Author Response · Authors · 2022-08-07
**Summary of Changes**

We thank the reviewers for the time and effort spent on providing valuable and constructive feedback which has truly improved the quality of our paper. We have uploaded a revised version of our paper, whose changes are marked in **magenta**. We would like to list the main updates we have made below.

1. [Section 1 and 2.1] Emphasized that we focus on the observational generalization of model-free RL algorithms.
2. [Section 2.2] Added more explanation of PPO and PPG for those who are not familiar with these algorithms.
3. [Section 4.3 and Figure 1] Stated the motivation of dynamics learning more clearly and used a more succinct expression to avoid misleading. Presented the network architecture of DDCPG.
4. [Section 5.2 and Table 1] Removed the highlights for individual games and rewritten the analysis of the experimental results.
5. [Section 6] Added more references on model-based RL algorithms and dynamics learning. Stated our contribution to dynamics learning more clearly.
6. [Appendix F.2] Added more ablation studies on dynamics learning (e.g. PPG + dynamics).
7. [Appendix G] Added the evaluation results using the RLiable library (https://github.com/google-research/rliable).

We hope that our updates have addressed all reviewers' questions and concerns. We would greatly appreciate it if the reviewers can check the updates and tell us if there are any further concerns.

Paper7811 Authors

---

### Meta-Review · Area_Chair_JeBM · 2022-08-27

**Recommendation:** Accept
**Confidence:** Certain

**Metareview:**

All reviewers were in favor of acceptance, and after reading the paper myself I am in agreement. The empirical results were good and the experimental work quite comprehensive. The method is well explained and the writing is clear and easy to read. The only real detraction I saw, distinct from things already mentioned by reviewers, was that there are some statements that feel overly strong given the presented results (e.g. L161, L179). I was curious about sensitivity to hyper-parameters, specifically the settings around how long each phase lasts, etc, along the lines of what was done in the PPG paper. That said, I would generally down-play the importance of this as the proposed method is using the same hyper-parameters as for PPG and appears to have undergone minimal hyper-parameter tuning.

In all I think this makes a clear contribution to the field and should be accepted.




**Award:**

No

---

### Decision · Program_Chairs · 2022-09-14

Accept